

**Geologic and geomorphic controls on rockfall hazard: how well do past rockfalls predict**
**future distributions?**
Josh Borella [1,2] , Mark Quigley [3,2], Zoe Krauss [4,1], Krystina Lincoln [5,1], Januka Attanayake [3],
Laura Stamp [5,1], Henry Lanman [6,1], Stephanie Levine [7,1], Sam Hampton [1,2], Darren Gravley [1,2]
[1] Frontiers Abroad, 3 Harbour View Terrace, Christchurch, 8082, New Zealand
[2] Department of Geological Sciences, University of Canterbury, Christchurch, 8041, New Zealand
[3] School of Earth Sciences, The University of Melbourne, Victoria, 3010, Australia
[4] Department of Geology, Colorado College, Colorado Springs, CO, 80903, USA
[5] Department of Geosciences, Williams College, Williamstown, MA, 01267, USA
[6] Department of Geology, Whitman College, Walla Walla, WA, 99362, USA
[7] Department of Geology, Carleton College, Northfield, MN, 55057, USA
Correspondence: Josh Borella (josh@frontiersabroad.com)
*KEYWORDS: Rockfall hazard, boulder spatial distributions, frequency-volume distributions,*
*Canterbury Earthquake Sequence, prehistoric rockfall boulders, deforestation, rockfall*
*source characteristics, rockfall physical properties, rockfall numerical modelling, rockfall,*
*Christchurch*
**Abstract**
To evaluate the geospatial hazard relationships between recent (contemporary) rockfalls and
their prehistoric predecessors, we compare the locations, physical characteristics, and
lithologies of rockfall boulders deposited during the 2010-2011 Canterbury earthquake
sequence (CES) (n=185) with those deposited prior to the CES (n=1093). Population ratios of
pre-CES to CES boulders at two study sites vary spatially from 5:1 to 8.5:1. This is interpreted
to reflect *(i)* variations in CES rockfall flux due to intra- and inter-event spatial differences in
ground motions (e.g., directionality) and associated variations in source cliff responses, *(ii)*
possible variations in the triggering mechanism(s), frequency, flux, record duration, boulder
size distributions, and post-depositional mobilization of pre-CES rockfalls relative to CES
rockfalls, and *(iii)* geological variations in the source cliffs of CES and pre-CES rockfalls. On
interfluves, CES boulders traveled approximately 100 to 350 m further downslope than
prehistoric (pre-CES) boulders, interpreted to reflect reduced resistance to CES rockfall
transport due to preceding anthropogenic hillslope de-vegetation. Volcanic breccia boulders
are more dimensionally equant, rounded, larger, and traveled further downslope than coherent
lava boulders, illustrating clear geological control on rockfall hazard. In valley bottoms, the
furthest-traveled pre-CES boulders are situated further downslope than CES boulders due to
*(i)* remobilization of pre-CES boulders by post-depositional processes such as debris flows,
and *(ii)* reduction of CES boulder velocities and travel distances by collisional impacts with
pre-CES boulders. A considered earth-systems approach is required when using preserved
distributions of rockfall deposits to predict the severity and extents of future rockfall events.


## 1 Introduction

Rockfall deposits pervade many mountainous and hilly regions worldwide (Varnes, 1978; Evans and Hungr, 1993; Wieczorek, 2002; Dorren, 2003; Guzzetti et al., 2003) and can provide important data for assessing future rockfall hazards (Porter and Orombelli, 1981; Keefer, 1984; Dussauge-Peisser et al., 2002; Copons and Vilaplana, 2008; Wieczorek et al., 2008; Stock et al., 2014; Borella et al., 2016a). Their characteristics (e.g. location, size, morphology) may be used to complement numerical rockfall modeling scenarios (Agliardi and Crosta, 2003; Dorren et al., 2004; Heron et al., 2014; Vick, 2015; Borella et al., 2016a) and inform engineering-design criteria for rockfall mitigation structures (e.g. impact fences, tiebacks, protection forests) (e.g. Agliardi and Crosta, 2003; Dorren et al., 2004; Guzzetti et al., 2004). However, natural and anthropogenic changes to the landscape (including changes to the rockfall source and slope areas) between successive rockfall events and the post-depositional history for rockfalls can be complex (e.g. Borella et al., 2016a,b). To better understand how past rockfalls provide suitable proxies for characterizing future hazard, comparisons between the geologic and geomorphic attributes of individual rockfall events and cumulative amalgamations of many events are valued. Critical evaluations of possible intervening changes to the landscape that may influence the mechanics of rockfall production and travel are an important component of these studies.

More than 7000 mapped individual rocks fell from cliffs in the Port Hills in southern Christchurch during the 2010-2011 Canterbury Earthquake Sequence (CES) in New Zealand's South Island (Massey et al., 2014). Most of the rockfalls (>6000) occurred during the 22 February 2011 moment magnitude (Mw) 6.2 and 13 June 2011 Mw 6.0 Christchurch earthquakes (Massey et al., 2014). Approximately 200 houses were impacted, 100 houses severely damaged, and five fatalities caused by falling rocks in the 2011 February earthquake (Massey et al., 2014; Grant et al., 2018). CES rockfalls were characterized by boulder-size distribution, runout distance (the distance a rock travels down a slope from its source), source-area dimensions, and boulder-production rates over a range of triggering peak ground accelerations (Massey et al., 2012a-e, 2014, 2017; Quigley and Mackey, 2014; Quigley et al., 2016).

Subsequent field investigations revealed an abundance of pre-CES rockfall deposits in CES rockfall areas (Townsend and Rosser, 2012; Mackey and Quigley, 2014; Borella et al.,




2016a,b), suggesting multiple rockfall events had occurred at these sites in the past (Mackey
and Quigley, 2014; Borella et al., 2016a,b; Sohbati et al., 2016). Retrospectively, these pre-
CES deposits had potential value to have contributed to hazard assessments during land-
planning and urban development in Christchurch prior to the CES; however, there is no
evidence that they did so (Townsend and Rosser, 2012; Litchfield et al., 2016). At one well-
studied location (Rapaki) in the Port Hills of southern Christchurch, CES and pre-CES boulder
populations were shown to have similar volumetric size and morphology characteristics, but a
significant population of CES boulders had longer maximum runout  distances than their pre-
CES counterparts (Borella et al., 2016a). Pre-CES rockfalls were dated using independent
approaches to >3-15 ka (Mackey and Quigley, 2014; Sohbati et al., 2016; Borella et al., 2016b).
With the aid of numerical modeling of rockfall trajectories (using RAMMS - rapid mass
movement simulation), these data were collectively interpreted to suggest that anthropogenic
deforestation between pre-CES and CES rockfalls was the primary cause for the observed
spatial distinctions in CES and pre-CES rockfall distributions (Borella et al., 2016a). Elsewhere
in the Port Hills and greater Banks Peninsula, the causes for differences in the spatial
distribution between CES and pre-CES rockfalls are less clear and in some locations the current
positions of pre-CES boulders extend further distances from source cliffs than their CES
counterparts. A more integrated and regional understanding of the geologic and geomorphic
controls on rockfall distributions has the potential to inform rockfall hazard analyses for land-
zoning and engineering considerations here and elsewhere (e.g. Lan et al., 2010).

In this study we document the location, volume, morphology, and lithology for individual
(n=1093) pre-CES rockfall boulders at two sites (Rapaki and Purau) in the Banks Peninsula
near Christchurch, New Zealand. The spatial distributions and physical attributes for pre-CES
boulders are compared to rockfall boulders (n=185) deposited at the same sites during the 2010-
2011 CES. RAMMS bare-earth and forested numerical modelling scenarios are conducted to
help evaluate the influence of geologic, geomorphic, and anthropogenic factors on rockfall
distributions, identify boulder sub-populations that have likely experienced post-emplacement
mobility, determine the relative timing of pre-existing rockfalls (i.e. prehistoric or historic),
and evaluate the efficacy of RAMMS in replicating empirical CES and prehistoric boulder
spatial distributions. We highlight the complexity of interpreting future rockfall hazard based
on former boulder distributions (particularly location) due to: (i) potential landscape changes
including deforestation, (ii) changes in rockfall source (e.g. progressive emergence of bedrock
sources from beneath sedimentary cover), (iii) remobilization of prior rockfalls by surface


processes including debris flows (primarily in channels), (iv) lithological variability effects on
the type of material liberated in successive events, (v) collisional impedance with pre-existing
boulders (particularly in channels/valleys), and (vi) variations in the location, size, and strong
ground motion characteristics of past rockfall-triggering earthquakes and their impact on
rockfall flux and boulder mobility.

**2 Geologic Setting**

**2.1 Overview**

Banks Peninsula, located on the east coast of New Zealand's South Island, is comprised of
three main volcanoes (Lyttelton, Akaroa, and Mt. Herbert) active between 11.0 and 5.8 Ma
(Hampton and Cole, 2009) (Fig. 1). The two study sites (Rapaki and Purau) located within
the inner crater rim of the Lyttelton Volcanic complex (Figs. 1, 2, 3), the oldest of the volcanic
centers and thought to be active from 11.0 to 9.7 Ma (Hampton and Cole, 2009). Source rock
at both sites is classified by Sewell (1988) and Sewell et al. (1992) as part of the Lyttelton
Volcanic Group (LVG) and consists of basaltic to trachytic lava flows interbedded with breccia
and tuff (Mvl). Numerous dikes and minor domes are observed within the LVG. Our field
observations support the reported lithologic descriptions for the two study locales. The inferred
strike and dip for lava flows nearest to the study sites indicates a shallow inclination in a
predominantly northerly direction for measurements nearest the Rapaki and Purau study sites
(Hampton and Cole, 2009). Sewell et al. (1992) reports a similar shallow northerly to
northwesterly dip of 12° for lava flows nearest Rapaki. The study areas were selected because
both have abundant pre-CES and CES rockfall boulders (Fig. 4) derived from lithologically
equivalent volcanic source rocks. Rapaki represents a case study location proximal to the
source of the 2011 February and June Christchurch earthquakes (epicenters ~2.5-5.0 km;
hypocenters ~ 5.6-7.0 km), while Purau is located more distally (epicenters ~6.6-8.4 km;
hypocenters ~8.9-10.3 km). Estimated rockfall-generating peak horizontal ground velocities
(PGV) at the Rapaki site in the February and June earthquakes were ≥ 30 cm s$^{-2}$ (Mackey and
Quigley, 2014).

**2.2 Rapaki study site**


The Rapaki study site is situated in the Port Hills of southern Christchurch (Figs. 1, 2) on the
southeastern slope of Mount Rapaki (*Te Poho o Tamatea*), which has a summit height of ~400
meters. The study hillslope is slightly concave to planar with a total area of ~0.21 km$^2$ and
faces to the east-southeast. The source zone consists of steep to subvertical bedrock cliffs
composed of stratified basaltic lava and indurated auto-breccia or pyroclastic flow deposits
(Fig. 5A-C). Breccia layers are thicker (~3-10 meters) and jointing is more widely spaced
(often >10 m). Coherent lava layers are comparably thin (<3 meters) and joints are more closely
spaced (generally <1-2 meter). Total height and length of the source rock are ~60 meters and
~300 meters, respectively (Fig. 5A). Below the source area, is a ~23°, grassy hillslope
composed of windblown sediment deposits (loess), loess and volcanic colluvium, and
overlying rockfall boulders (both CES and pre-CES) (Bell and Trangmar, 1987). Rapaki village
(estimated population=100 residents) lies at the hillslope base at elevations of ~70 meters (asl)
to sea level (Figs. 3, 4). Anthropogenic deforestation has exposed a hillslope that is currently
experiencing accelerated erosion (Borella et al., 2016a,b) in the form of mass wasting and
tunnel gully formation. Shallow landslides, including debris and earth flows, are most prevalent
in upper to mid-slope positions, while rill and gulley erosion predominate in lower slope
positions.

Rockfall is a dominant surface feature at the Rapaki study site (Mackey and Quigley, 2014;
Vick, 2015; Borella et al., 2016a,b). Pre-CES and CES rockfall boulders at the study site can
be divided into two dominant lithology types: volcanic breccia (VB) and coherent lava (CL)
basalt. During the 22 February and 13 June 2011 earthquakes, more than 650 individual CES
boulders ranging in diameter from <15 cm to >3m were dislodged from the volcanic source
rock near the top of Mount Rapaki, many impacting and destroying residential homes (Massey
et al., 2014; Mackey and Quigley, 2014).

**2.3 Purau study site**

Purau is located on the southern side of Lyttelton Harbour, approximately 5 kilometers
southeast of Rapaki (Figs. 1, 3). Slopes at Purau have a west-northwest aspect, the opposite of
the Rapaki study hillslope. Mapping of pre-CES and CES rockfall was performed on and within
several interfluves (spurs) and bounding valleys, respectively (Fig. 3) and encompassed a total
area of ~1.4 km$^2$. The source rock geology at Purau, including lithology and structure, is
equivalent to that observed at Rapaki (Fig. 5D,E). The ridgeline (i.e. volcanic source rock) to



the east obtains a maximum elevation of ~440 meters. Locally, individual vertical to subvertical
bluff faces are estimated to be ~20-30 meters in height. From the base of the volcanic source
rock, slopes extend downward toward Purau Bay at angles ranging from ~30° to ~5° near Camp
Bay Road (Fig. 3). Field observations indicate the volcanic rock is overlain by loess, loess- and
volcanic-colluvium, and pre-CES and CES rockfall boulders of small (e.g. <1 m$^3$) to extremely
large size (e.g. >100 m$^3$). Deforestation of Purau slopes has left the hillside covered primarily
in low-lying grass and bush. Shallow slips are abundant and are commonly observed on steep
slopes, including valley flanks. Maximum landslide depth is typically ~1-1.5 meters and often
exposes volcanic bedrock at bottom, indicating the overlying sediment is relatively thin. Tunnel
gulley erosion predominates on canyon flanks and at lower elevations.

**3 Methods**

**3.1 Field mapping and characterization of CES and pre-CES rockfall boulders**

We mapped 1276 individual rockfall boulders at the Rapaki (pre-CES=408; CES=48) and
Purau (pre-CES=684; CES=136) study sites for boulder volume ≥1.0 m$^3$ (see Supplementary
Data, Tables S1-S4). Where safety conditions permitted, pre-CES and CES rockfall boulders
were mapped to the base of the volcanic source rock. Location (latitude/longitude) and
elevation (meters above sea level) were recorded for each rockfall deposit using a hand-held
Garmin GPSMap 62s device. Boulder dimensions (i.e. height, length, width) were tape
measured in the field. For pre-CES boulders partially buried to the degree that only two
dimensions were adequately measurable, the shorter of the two measured lengths was used for
the 3$^{rd}$ dimension, thus insuring a conservative boulder size estimate. No rounding factor was
applied to volumetric estimations of pre-CES boulders. The lithology type was determined for
each pre-CES boulder and was based primarily upon the observed dominant rock 'texture'.
Boulder lithologies were categorized as VB or CL. Transitional lithologies were rarely
observed (<1% of total) and assigned as VB or CL based on the volumetrically predominant
rock type.

**3.2 Boulder runout distance**


Boulder runout distance was determined by measuring the shortest horizontal and ground-
length distances, perpendicular to slope contour lines, from the nearest potential bedrock source
areas to mapped boulder locations using Google Earth Professional (see Supplementary Data,
Tables S5-S8). Runout distance was calculated for 409 pre-CES boulders and 48 CES boulders
(for volume $\geq 1.0$ m$^3$) at Rapaki. Due to safety concerns we were unable to record locations for
pre-CES boulders within ~100 meters (map-length) of the volcanic source rock at this site.
However, boulder frequency counts (for boulder volume $\geq 0.1$ m$^3$) were field measured within
a 300 m$^2$ area at distances of 0-10 meters (n=31), 30-40 meters (n=35), 60-70 meters (n=77),
and 100-110 (n=24) meters from the volcanic source rock (see Appendix 1, Fig. A1). The
boulder frequency counts at these distances were used to extrapolate the number of boulders
across remaining sections of the study site, consistent with visual inspection of air photos. At
Purau, four separate geomorphic domains (PD1-PD4) were created to evaluate pre-CES and
CES boulder runout distance (see Fig. 3; Supplementary Tables S7, S8). The domains include
interfluve and valley morphologies and target areas with both CES and pre-CES rockfall
boulders, and cases where the pre-CES rockfalls were sourced from a single or limited number
of rock exposures. We generally report map-length runout distance within this paper.

We used the empirical shadow angle method (Lied, 1977; Evans and Hungr, 1993) to analyze
the travel distance of rockfalls at Rapaki and Purau. The shadow angle is the arctangent of the
relationship Ht/Lt, where Ht is the height of fall on the talus slope (elevation difference between
the apex of the talus slope and final emplacement location of the rockfall block) and Lt is the
travel distance on the talus slope (horizontal distance between the apex of the talus slope and
the final emplacement location of the rockfall block) (see Copons, 2009; Lied, 1977; Evans
and Hungr, 1993) (see Appendix 1, Fig. A2). The shadow angle method is most suitable for
our study (compared to the reach or 'Fahrboschung' angle) because it does not require
identifying the source release location for individual rockfall blocks, a parameter we are unable
determine for the pre-CES and CES rockfalls.

**3.3 RAMMS rockfall modeling**

Three model scenarios were conducted using the Rapid Mass Movements System (RAMMS)
software (Bartelt et al., 2013; Leine et al., 2014). RAMMS_1 represents a bare-earth CES
model and was performed to test the reliability of RAMMS in replicating the spatial



distribution for CES rockfalls at Purau. RAMMS_2 assumes a vegetated slope and simulates
hillslope conditions prior to deforestation (i.e. prehistoric). RAMMS_3 models the potential
future rockfall hazard at Purau and assumes a bare-earth (deforested) hillslope and dry soil
moisture conditions to insure a worst-case (conservative) outcome. Please see Supp.
Information for more detail on the individual RAMMS modeling scenarios.

The Purau terrain was modelled using a 4-m DEM (digital elevation model) derived from
LIDAR (light detection and ranging) surveys to model CES (bare-earth scenario) and pre-CES
prehistoric (forested slope scenario) rockfall distributions. The rockfall boulders were
modelled as rigid polyhedral. The source areas (i.e. volcanic rock) and remaining runout terrain
types (i.e. loess and loess/volcanic colluvium) (Appendix 2, Table A1 and Figs. A1-A3) for
the RAMMS model scenarios (i.e. RAMMS_1, _2, _3) were chosen following the methods of
Vick (2015) and Borella et al. (2016a) and delineated as polyline (Appendix 2, Figs. A2, A3)
and polygon shapefiles (Appendix 2, Fig. A3) in ArcGIS from field observations, desktop study
of orthophotography, and satellite imagery.

Boulder shape and size are highly influential in the dynamics and runout of a rockfall event
(e.g. Leine et al., 2014; Latham et al., 2008). Boulder shapes and sizes used in the model
simulations were representative of the true boulder geometries observed at Purau and Rapaki
(Borella et al., 2016a). Rocks shapes were created using the RAMMS 'rock builder' tool, which
creates boulder point clouds based on a user-defined shape and size. All boulder shapes
reflected 'real' rock bodies that have been field-scanned. For each size class of boulder, varying
shapes were selected, which are simplified to equant, flat, and long. Please see Supp.
Information for more detail on boulder shape and size distributions utilized in each of the
RAMMS modeling scenarios.

Vegetation was modelled in RAMMS as forest drag, a resisting force acting on the rock's
center of mass when located below the drag layer height. The forest was parameterized by a
drag coefficient, effective up to the input height of the vegetation layer. Typical values for the
drag coefficient range between 100 and 10,000 kg/s (Bartelt et al., 2013; Leine et al., 2014).
Vegetation was assigned an effective height of 10 m. A variable forest density was applied to
account for the presumed denser vegetation (on average) within drainage valleys at the Purau
study site (Appendix 2, Fig. A4). We assume more surface and subsurface water would be
focused into topographic lows and would therefore promote denser tree growth. Within



drainage valleys a uniform drag force of 6000 kg/s was applied to each of the simulated
boulders. Elsewhere at the study site, a drag force of 3000 kg/s was applied. These forest values
are equivalent to those utilized in Borella et al. (2016a) at Rapaki in the Port Hills of southern
Christchurch. We also simulated a uniform forest density increase of 10000 kg/s (see Results).
As evidenced by modern native forest analogs, tree growth was extended upward to the base
of the source rock and was also applied to areas between outcropping volcanic source rock.

**3.4 Strong ground motions near rockfall source cliffs**

Strong ground motion accelerograms for stations LPCC, D13C, D15C, and GODS were
obtained from GeoNet (www.geonet.org.nz/, Fig. 6) to analyze the influence of ground motion
on rockfalls. All these stations are Kinematrics Etna instruments except LPCC, which is a
CUSP-3 instrument. LPCC recorded both Mw 6.2 event on 2011-02-21 and Mw 6 event on
2011-06-13. The other stations were installed following the Mw 6.2 earthquake and thus
recorded only the Mw 6 earthquake. The data were sampled at 0.005 s (Nyquist frequency 100
Hz) and filtered with an effective passband having corners ~0.05 Hz and ~40 Hz. We integrated
accelerograms to produce velocity seismograms and computed envelopes using ENV = sqrt[
$x(t)^2 + H(x(t))^2$ ], where x(t) are time points in the seismogram and $H$ is the Hilbert
transform. The particle velocity hodograms are calculated in the horizontal plane by rotating
the horizontal orthogonal components of the seismogram to a standard N-S E-W coordinate
system. The time window of particle velocity hodograms is ± 5 s around the peak of the
envelope of the east component. This ensures that the most significant ground motion resulting
from both phase and group velocity peaks is accurately captured. Following a similar
procedure, we computed particle motion hodograms by integrating accelerograms twice. These
are given in Fig. 7 (A-E). Additional methods were used to analyse D13C data following
interpretation of initial results; these are described in  section 5.8.

**4 Results**

**4.1 Rockfall mapping and boulder frequencies**

**4.1.1 Rapaki**


A comparison of the spatial distributions for pre-CES and CES rockfalls at Rapaki (Fig. 2)
indicates that pre-CES rockfalls are more concentrated near the source area and have shorter
maximum runout distances (560±15 m) compared with the furthest travelled CES rockfalls
(700±15 m), which impacted the Rapaki village during the 2011 Christchurch earthquakes. The
CES rockfalls represent a subset of the pre-CES rockfall data set; the ratio of pre-CES (n=409)
to CES (n=49) rockfalls at Rapaki is ~8.5:1 (Fig. 2). The pre-CES and CES rockfall data sets
are separated into VB and CL boulders (Fig. 2, 4) to understand the influence of volcanic
lithology on rockfall runout and final resting location. Very few CL boulders with volume ≥1.0
$m^3$ exist for pre-CES (n=18) and CES (n=3) rockfalls at Rapaki. Pre-CES and CES VB boulders
display longer average and maximum runout distances than their CL counterparts (Fig. 2), and
CES CL and VB boulders display longer average and maximum runout distances compared
with their pre-CES equivalents. The ratio of pre-CES VB to CL and CES VB to CL rockfall
boulders is ~22:1 and ~15:1, respectively (Fig. 2).

**4.1.2  Purau**

Pre-CES and CES rockfalls are widely distributed at the Purau study location (Fig. 3). Rockfall
boulders are deposited on interfluves but are predominantly concentrated within nearby
canyons, highlighting the strong influence of topography at the site (Fig. 3). Seven (7) CES
detachment zones were identified in the field. CES rockfall boulders nearest to the Purau
village display the longest runout distance (372 m) and most distinct spatial contrast with
similarly sourced pre-CES boulders (deposited within ~105 meters of the local volcanic source
rock) (Fig. 3A). Elsewhere, pre-CES boulders can be observed at further distances from the
source rock than CES rockfalls. The ratio of pre-CES to CES rockfall boulders is ~5:1 (Fig.
3A). Pre-CES VB boulders are deposited throughout the Purau location, while the deposition
of CL pre-CES boulders is concentrated within the central and southern drainage canyons (Fig.
6A). The ratio of pre-CES VB to CL boulders is ~2:1 (Fig. 3B). CES VB boulders (n=127)
significantly outnumber CL boulders (n=9) at the Purau site (Fig. 3C), reflecting the lack of
detachment within CL source rock lithologies during the CES. The ratio of CES VB to CL
rockfall boulders is ~14:1 and represents a significance difference compared with the
corresponding pre-CES VB:CL ratio (Fig. 3C).

**4.2  Boulder morphology and other characteristics**






VB boulders (Fig. 4A-F) contain small to large porphyritic volcanic clasts that exhibit minor
to moderate vesicularity (up to ~10%) and are embedded within a finer crystalline and ash-
bearing matrix (see Fig. 4A,C,D,F). They are dominated by equant (all axes equal length)
shapes (see Fig.4C) although elongate (two short axes, one long) forms are observed. Flat (one
short, two long axes) morphologies are rare. VB pre-CES boulder surfaces show a high degree
of weathering and surface roughness (Fig. 4A-D,F). The surface roughness results from in-situ
differential weathering between the finer crystalline host matrix and more resistant embedded
volcanic clasts (see Fig. 4D).  Surfaces show deep pitting, with amplitudes often exceeding 5-
10 centimeters in height. CL boulders (Fig. 4G-K) are more texturally homogenous, contain
fewer vesicles (estimated ~<1%) and exhibit a higher relative density (Carey et al., 2014;
Mukhtar, 2014). The pre-CES CL boulder surfaces exhibit low surface roughness (i.e. smooth
compared with VB boulders). Elongate and flat boulder morphologies predominate for CL
boulder lithologies (Fig. 4G-K).

Both VB and CL pre-CES boulders can be observed partially to nearly completely buried by
loess-colluvium (see Fig. 4A,B,G). Instances do occur, however, where no sediment is built-
up at the boulder backside (Fig. 4C) due to erosion (including tunnel gully formation). Burial
in hillslope sediment is most common for boulders located on midslope and footslope positions,
rather than those located on upper slope elevations, where erosion dominates. Pre-CES
boulders located in drainage canyons are subject to rapid deposition and erosion, and therefore
can be found without any sediment pile-up or preserving large colluvial wedges. VB boulders
preserve the thickest colluvial wedge sediments (see Fig. 4B).

**4.3  Source rock characteristics**

The volcanic source rock at Rapaki (Fig. 5A-C) and Purau (Fig. 5D,E) is comprised of
interlayered VB and CL layers (Fig. 5A-E). The breccia layers comprise the bottom and top of
discrete lava flows, while the coherent lava generally occupies the center of the lava flow where
cooling was not as rapid and there was less interaction with the substrate and/or cooling
interface (Fig. 5C-G). Jointing is pervasive within the volcanic source rock, but to varying
degree depending upon layer composition and corresponding texture. Layers comprised of CL
exhibit the highest fracture density (Fig. 5E,F) and were formed during primary cooling of the
lava flow, producing a columnar-style pattern. The CL layers contain numerous intersecting





sub vertical to vertical, to curvilinear joint sets, with spacing rarely exceeding ~1-2 meter. The
small joint spacing imparts a first-order control on CL boulder size and is reflected in the small
size range for pre-CES CL boulders. Layers comprised of VB exhibit a lower fracture density,
with joints more widely spaced (and irregular in shape), often 5-10 meters or greater apart (Fig.
5D,E). The wider spacing for joints within VB layers promotes greater rockfall boulder volume
(see Section 4.4. below).

During the CES, rockfall detachment occurred within approximately 9% (by area) of the
volcanic source rock overlying the Rapaki study hillslope (Fig. 5A). The volcanic source rock
is comprised of 86% VB and 14% CL. 69% of the detachment areas occurred within VB and
the remaining 31% within CL (Fig. 5A). However, 20% of the identified CL source rock
detached during the CES, while only 7% of the identified VB source rock detached during the
CES, indicating the CL lithology is more susceptible to detachment. Due to its significant size
and safety concerns, a similar characterization was not performed for the Purau volcanic source
rock.

**4.4 Boulder volume**

The size and frequency-volume distributions for pre-CES and CES rockfall boulders (for
volume $\geq 1.0$ m$^3$) at Rapaki and Purau display similarity (Fig. 8A,C) and can be modeled using
power law functions (Fig. 8B,D), with the number of rockfall boulders decreasing significantly
as volume increases. Overall, statistical coherence is observed at the 25$^{th}$, median, and 75$^{th}$
percentile boulder sizes; however, pre-CES rockfalls are consistently higher for each of the
size categories at the two study locations (Table 1). Rapaki displays the highest pre-CES to
CES variance for 25th, median, and 75$^{th}$ percentiles, while Purau records the biggest pre-CES
to CES variance for the average, 95$^{th}$ percentile, and maximum boulder volumes (Table 1, Figs.
8A,C).

At Rapaki, VB pre-CES and CES boulder volumes display a similar trend (Fig. 8E) compared
to the pre-CES and CES boulders (see Fig. 8A), indicating the dominance of VB boulders for
volume $\geq 1.0$ m$^3$. Pre-CES VB boulders display higher volumes in each of the size categories,
particularly for median and maximum boulder sizes (Table 2). Pre-CES CL boulders display
consistently higher values for each of the size categories with the exception of the 75$^{th}$



percentile (Fig. 8E, Table 2). At Purau, CES VB and CL boulders exhibit a smaller distribution
of boulder sizes compared with their pre-CES equivalents (see Fig. 8F). Pre-CES VB and CL
boulders are higher in each of the size categories (Table 2, Fig. 8F), with the exception of the
median boulder size, where the CES CL median boulder volume is slightly more than the pre-
CES CL value (Table 2). It is notable that the highest percent (%) variance in boulder volume
between pre-CES and CES boulders is recorded with the Purau VB boulders (Table 2); the
only exception is for maximum boulder size, where the percent (%) difference between Purau
CL pre-CES and CES boulders is even greater (Table 2).

The volume and frequency ratios for pre-CES and CES rockfall boulders are plotted in Figure
9A. The pre-CES to CES boulder volume ratios at Rapaki and Purau range from ~8-12 and ~7-
37, respectively (Table 3A, Fig. 9A). The corresponding frequency ratios are consistently
lower, ranging from ~6-8.5 and ~3.5-27.5 (Table 3A, Fig. 9A). Overall, the boulder volume
and frequency ratios are greater at Rapaki, with the exception of the CL lithology (Tables 3B,
3A, and Fig. 9A).

The calculation of VB and CL boulder percentages at Rapaki for pre-CES and CES rockfalls
indicates that VB boulders comprise ≥ 98% by volume and ≥ 94% by frequency (n) for all
Rapaki conditions, while at Purau the corresponding percentages are ≥ 90% (volume) and
≥64% (frequency) respectively (Table 3B). All of the lowest VB percentages exist at the Purau
study location (see Table 3B, individual domain data).

**4.5 Boulder runout distance**

The frequency-runout distance distribution for pre-CES boulders at Rapaki can be
characterized by power and exponential laws (Fig. 9B), with the number of rockfall boulders
with long runout distances decreasing dramatically with increasing distance from the volcanic
source rock. The exponential regression is best fit to the entire data set (including extrapolated
boulders within 100 m of source rock), while the power law displays the strongest fit for the
mapped rockfall boulders (Fig. 9B). CES rockfalls display a poor exponential fit and do not
indicate a similar inverse relationship between boulder frequency and runout distance (Fig.
9B). The frequency-runout distribution for CES rockfalls indicates that the number of boulders
remains more or less consistent regardless of distance from the source rock. Using the shadow



angle method we plot travel distance on the talus slope (Lt) versus height on the talus slope
(Ht) with a fitted polynomial regression line (Fig. 9C). The correlation coefficient is 0.9699 for
CES rockfalls and 0.9717 for pre-CES rockfalls (Fig. 9C). The minimum shadow angle for
pre-CES is 25°, while the minimum shadow angle (for the furthest traveled CES rockfall
boulders) is 23°. At Rapaki, the maximum runout distance for pre-CES and CES VB boulders
exceeds the furthest travel distances for pre-CES and CES CL boulders, respectively (Table 4).
The CES VB boulders exceed pre-CES VB runout by ~165 meters and CES CL boulders
exceed CL pre-CES runout by ~138 meters (Fig. 2A,B; Table 4).

At Purau, Lt versus Ht is plotted for four (4) separate geomorphic domains (PD1-PD4) to
evaluate the distribution of pre-CES and CES boulder runout distances (Fig. 9D; see Fig. 3 for
domain locations). The pre-CES and CES rockfalls for the individual domain data sets are
characterized by a variety of regression functions with high correlation coefficients (Fig. 9D;
Supplementary Data, S24). CES rockfalls in PD1 and PD4 have significantly further maximum
runout distances than their pre-CES counterparts, while the inverse is evident in PD2 and PD3.
[We note that only two CES boulders were observed in PD2.] The minimum shadow angle for
pre-CES rockfalls at Purau is 25°, while the corresponding minimum CES rockfall shadow
angle is 18°. At Purau, the longest recorded runout distances occur for pre-CES CL and VB
boulders and CES VB rockfall boulders within PD3 (Table 4).

At Rapaki, no relationship has been obtained plotting individual boulder volumes and the
tangent of the shadow angle (Fig. 9E). A wide range of boulder sizes are evident for the full
spectrum of pre-CES and CES rockfall runout distances by means of the shadow angle. The
same is largely true at Purau, where correlations for the individual domains (PD1-PD4) are
poor and the data has a high degree of scatter (i.e. low correlation coefficients); although the
data does show a slight negative relationship between block volume and $Ht/Lt$ ratio value (that
is, a slight increase in runout distance as boulder size increases) (Fig. 9F).

### 4.6   RAMMS rockfall modelling


### 4.6.1   RAMMS_1


Final resting locations (n=1072) are generated for simulated rockfalls released from the seven
(7) field-identified CES detachment zones at Purau (labeled CES-1 through CES-7) (Fig. 10A).


The empirical CES boulder locations are depicted as red circles. RAMMS_1 (bare-earth CES
model scenario) is successful in replicating the overall spatial pattern for detached and
distributed CES rockfalls at Purau for locations CES-3, -4, -5, -6, and -7. Below the CES-7
source rock, RAMMS maximum runout distances (~370 m) are well matched to the maximum
travel distance for mapped CES rockfalls (~357 m). Maximum runout distances for the
RAMMS boulders are overestimated at CES-1 and CES-2 (Fig. 10A). We note that only 2
boulders were released at CES-1 during the CES and were deposited within ~12 meters of the
source rock. RAMMS_1 effectively captures the lateral dispersion for the mapped CES
boulders at CES-2, CES-3, and CES-4, but overestimates this effect within the CES-5 and CES-
6 valleys, and slightly underestimates the lateral dispersion of CES rockfalls beneath CES-7.

**4.6.2 RAMMS_2**

The RAMMS_2 model scenario (forested hillslope) is moderately successful (slight
overprediction) in replicating the overall spatial distribution and maximum runout distances
for the majority of mapped pre-CES rockfalls at Purau (Fig. 10B). The exception is area CES-
7, where RAMMS predicts deposition of pre-CES boulders significantly farther (~325 m) from
the source rock than is evident in the field (~80 m). Elsewhere, the greatest variance in
maximum runout distance between RAMMS_2 and the mapped pre-CES boulders is ~75-100
m (see Fig. 10B). An increase in forest density to 10,000 kg/s, spread uniformly across the
study site produces the best fit to the pre-CES boulder spatial distributions (in particular,
maximum runout distance) (see Figure 10B, white dashed line). RAMMS_2 successfully
models the lateral dispersion for the mapped pre-CES boulders (with the exception of area
CES-7) (Fig. 10B). The RAMMS_2 model scenarios identify pre-CES rockfall boulders that
have likely experience post-emplacement mobility (see Fig. 10B). Note the collection of pre-
CES boulders within the central drainage canyon that exceed the limit of simulated RAMMS
boulders (Fig. 10B). Field observations confirm that boulder depositional patterns beyond the
limits of the final resting locations for RAMMS simulated rockfall boulders are consistent with
deposition by debris flow and other transport/deposition processes. Importantly, we observe no
mapped pre-CES boulders outside of the valleys that exceed the RAMMS_2 simulated
maximum runout distances.

**4.6.3 RAMMS_3**



RAMMS_3 models the potential future rockfall hazard at Purau and assumes a bare-earth
(deforested) hillslope and dry soil moisture conditions to insure a worst-case (conservative)
outcome (Fig. 10C). As expected, RAMMS_3 rockfalls obtain higher kinetic energy, velocity,
and jump heights than RAMMS_2 boulders (see Supplementary Data, S18, S19), and as a
result, runout farther than the RAMMS_2 boulders (Fig. 10B). On average, maximum runout
distance for RAMMS_3 boulders is ~450-500 m, representing an increase of ~100-150 m
compared with RAMMS_2 boulders, a difference consistent with results from RAMMS
numerical modeling at Rapaki (see Borella et al., 2016a). The RAMMS_3 results indicate that
the existing residence furthest to the north (S1) (Fig. 10C) and potential development at S2
could be adversely impacted by future rockfall events. With the exception of area CES-7,
RAMMS_3 maximum runout distances are well in exceedance of the mapped locations for the
CES rockfall boulders (Figs. 10A,C) and highlights the potential input from additional
detachment sites within the Purau volcanic source rock.

**4.7  Strong ground motion data**

High frequency data show complex velocity and displacement paths for any given site. The
variations across the sites are significant, and they have been reported previously (Van Houtte
et al., 2012; Bradley, 2016). Even for the same site (LPCC, Fig. 7A,B), particle velocity and
motion hodograms show different polarization characteristics for different earthquakes. Peak
velocities and displacements recorded at LPCC site are higher for the Mw 6.2 than the smaller
event Mw 6.0 (Fig 7A, B). The observed inter-site and inter-event variations in polarization of
peak velocities and displacements can be attributed to source radiation pattern (Lee, 2017) and
complex wave propagation effects such as scattering. For instance, simulating high frequency
(> 1 Hz) 3-D wavefields, Takemura et al. (2015) showed that near-station irregular topography
amplifies scattering of seismic wavefield, producing long coda and distortions to P wave
polarizations. This is not surprising given that Fresnel volume – the region to which a
transmitting seismic wave is sensitive – is inversely related to wave frequency (Spetzler and
Snieder, 2004), due to which near-station geological conditions modify wave characteristics at
high frequencies. The control of near-station geology over polarization and amplification
characteristics at high frequencies (Bouchon & Barker, 1996) reduces our ability to extrapolate
these characteristics to distant sites.

**5  Discussion**






**5.1 Rockfall spatial distributions and frequencies**


At Rapaki, significant differences in spatial distribution between the pre-CES and CES boulder
populations are observed (Fig. 2 and Table 4). The increased distance for the CES rockfall
boulders is interpreted as an effect of anthropogenic deforestation on the hosting hillslope,
which enabled CES boulders to travel further than their pre-CES counterparts due to reduced
resistance from vegetation (Borella et al., 2016a). The increase in CES runout distance
(~165±15 m) (and corresponding reduction in minimum shadow angle) resulted in significant
impact and damage to homes and infrastructure in the Rapaki village, and highlights the
importance of considering the effects that modifications to hillslopes may have on rockfall
hazard. At Rapaki, pre-CES VB boulders are present in significantly greater number and have
further average and maximum runout distances than the pre-CES CL boulder lithologies (Fig.
2A, Table 4). A similar relationship is evident between the CES VB and CL boulders, where
CES boulders with the furthest runout distances are exclusively comprised of volcanic breccia
(Fig. 2B). It is possible that the reduced runout distances for pre-CES and CES CL boulders is
a statistical counting bias (i.e. low number of CL boulders for volume ≥1.0 m³), but a more
plausible explanation is that the reduced runout distance for CL boulder lithologies is a result
of CL boulder shapes being dominated by elongate and flat morphologies (Fig. 10A-F), which
would have more difficulty traveling downslope.

At Purau, discerning the differences in spatial distribution between pre-CES and CES rockfalls
is more difficult, primarily due to the topographic forcing of rockfalls into nearby drainage
valleys and post-emplacement mobilization (Fig. 3). Location CES-7 (furthest southern
rockfalls) does show a similar pre-CES:CES spatial scenario to Rapaki, with CES boulders
traveling significantly further than their pre-CES equivalents (see Fig. 5); a discrepancy which
could also be attributed to intervening deforestation on the hillslope. However, elsewhere at
the Purau field site inverse spatial scenarios are evident, with pre-CES boulders deposited
further from the source rock than their CES counterparts (see Fig. 2A, Table 4). This is
primarily observed within drainage valleys where field observations suggest pre-CES boulders
have been remobilized (debris flows, floods) and carried further from the source rock following
their initial emplacement.



The CES rockfall boulders at both sites represent a subset of the larger pre-CES rockfall
database, suggesting the preservation of multiple pre-CES rockfall events. The ratio for the
number of pre-CES to CES rockfall boulders is higher at Rapaki (~8.5:1) than Purau (~5:1)
(Table 3, Figs. 2, 3). One cause of the observed difference may be the higher number of CL
boulders with size ≥1.0 m$^3$ at the Purau study site (Fig. 8E,F). At Rapaki, most of the
detachment within the CL source rock generated boulder volumes below the 1.0 m$^3$ threshold.
As a result, the ratio of pre-CES VB:CL boulders is significantly higher at Rapaki (~22:1)
(Table 3B, Fig. 2A) than Purau (~2:1) (Table 3B, Fig. 3B). This contrasts with the ratio of CES
VB:CL boulders at Rapaki (~15:1) (Table 3B, Fig. 2B) which shows near equivalence to Purau
(~14:1) (Fig. 3C). The CES VB:CL ratio at Purau is more consistent with our field observations
where VB predominates in the source rock. Overall, the results indicate there is a high degree
of lithologic variation and discontinuity spacing (e.g. joints) within the source rock and
suggests the cumulative ratio of VB:CL boulders can be significantly different from that
generated locally during a single rockfall event.

**5.2 Boulder morphology and other characteristics**

It is well-established that boulder morphology (shape) plays a primary role in the spatial
distribution of the rockfalls (e.g. Leine et al., 2014). The shapes for the VB (Fig. 4A-E) and
CL (Fig. 4G-K) boulders are primarily controlled by pre-existing discontinuities in the source
rock; in particular, jointing. We modeled the influence of boulder shape on spatial distribution
for the VB and CL lithologies assuming detachment from the CES-7 site (under bare-earth
conditions) using RAMMS (Fig. 11). To eliminate the effect of boulder size, a volume of 1.0
m$^3$ was assumed for all rockfall boulders. The VB boulders were assigned a range of equant
boulder shapes, while CL boulders were assigned only elongate and flat boulder morphologies.
The model results highlight the differences in boulder spatial distribution resulting from
differences in boulder shape, with equant (VB) boulder lithologies displaying a significantly
higher relative percentage of longer runout distances (Fig. 11A) compared with the
elongate/flat (CL) boulder morphologies (Fig. 11B). We recognize that the modeling represents
an ideal scenario (i.e. other transition morphologies do exist for the VB and CL boulders) and
was conducted primarily to provide a sense for the expected spatial patterns assuming the
distinct VB and CL boulder shapes. Further work is required to verify coherence between field
observations and model results.



### 5.3  Source rock characteristics



We combined high-resolution aerial photography (from UAV) with field observations to
characterize the Rapaki source rock. The volcanic source rock is comprised of 86% VB and
14% CL (VB:CL ratio=~6:1) (Fig. 5A-C) by percent area, values that are lower than the
corresponding VB and CL percentages determined from rockfall frequency and volume for the
pre-CES (96% VB and 4% CL) and CES (94% VB and 6% CL) rockfalls. We attribute the
percent differences between source rock and rockfalls to the influence of the larger VB boulder
sizes and the lower number of CL rockfalls meeting the $\geq 1.0$ m$^3$ size threshold. These two
factors also explain detachment during the CES, where 69% of the detachment areas occurred
within VB and the remaining 31% within CL (Fig. 5A-C), yielding a lower VB:CL ratio of
~2:1 compared with the corresponding boulder volume and frequency ratios (~15:1 and ~52:1,
respectively) (Table 3B). Comparisons between volcanic source rock characteristics and
boulder volumes (VB and CL) are discussed in Section 5.4. (see below).

We were unable to conduct a similar source rock investigation at Purau because the size of the
source rock was too great and in several cases deposition of rockfall boulders into discrete
geomorphic domains resulted from detachment on multiple source rock outcrops. However,
observations were made for the Purau source rock (Fig. 5D,E) as well as other volcanic coastal
cliff outcrops at Sumner (Fig. 5F) and Red Cliffs (Fig. 5G). Field observations indicate CL
layers at Purau are not as prevalent as (and generally thinner than) VB layers, but in some cases
may exceed a thickness of 5 meters, which is thicker than CL layers observed at Rapaki (see
Fig. 5B,C). At Sumner and Redcliffs, VB and CL layers display roughly equivalent thicknesses
(~2-3 m), a condition not apparent at Rapaki or Purau. The variability in layer thickness
presumably reflects differences in proximity to source vents and differing conditions during
primary cooling of the lava flows.

### 5.4  Boulder volume



The size and frequency-volume distributions for pre-CES and CES rockfalls at Rapaki (Fig.
8A,B) and Purau (Fig. 8C,D) can be modeled using a power law and indicate a predictable
decrease in the number of boulders as boulder volume increases. A power law frequency-size


distribution is well-established (e.g. Dussauge-Peisser et al., 2002; Guzzetti et al., 2002) for
rockfalls globally and has also been successfully applied for CES rockfalls in Banks Peninsula
(Massey et al., 2014). At both study locations, pre-CES rockfalls exceed the size of their CES
counterparts in all statistical categories (Table 1). The net increase in volume distribution for
pre-CES boulders could represent a statistical effect and reflect the inclusion of more boulders
into the rockfall data set through time (which would increase the likelihood of more large
boulders) and/or could reflect higher shaking intensities and/or source rock vulnerability during
pre-CES events.

A comparison of rockfall volumes between the two sites indicates that pre-CES rockfalls at
Rapaki are greater for the $25^{th}$, median, and $75^{th}$ percentile sizes (Table 1) while Purau exhibits
larger sizes for the $95^{th}$ percentile, maximum, and mean boulder categories (Table 1). For CES
boulders, the $25^{th}$, median, $75^{th}$, and $95^{th}$ percentile Rapaki CES boulders are slightly larger
compared with Purau CES boulders, while the maximum and mean boulder size categories are
higher at Purau (Table 1). Although differences are evident, the overall size distributions are
comparable (Table 1). Variations in CES vs. pre-CES boulder volumetric distributions for the
same lithologies could reflect structural and/or more subtle lithologic variability within the
source cliffs from which boulders were derived, and/or post-detachment weathering during
boulder transport or *in situ*.

The volume for pre-CES and CES VB boulders is significantly larger than the corresponding
CL boulders at Rapaki (Fig. 8E, Table 2) and Purau (Fig 8F, Table 2), reflecting the
predominance of VB within the source rock and wider joint spacing within the thicker VB
layers. As expected, the pre-CES VB and CL boulder sizes exceed those of their CES
equivalents, with the exception of the $75^{th}$ percentile CL boulders at Rapaki and median CL
boulders at Purau (Table 2, Figs. 8E,F). It is notable that the largest percent variance between
pre-CES and CES boulder size occurs for the Purau VB boulders (with the exception of
maximum boulder size) (Table 2). We are uncertain why this difference is greatest within the
Purau VB boulders, but could reflect a smaller joint spacing at the CES VB detachment sites.

**5.5 Boulder runout distance**

The frequency-runout distance distribution for pre-CES boulders at Rapaki can be modeled
using a power law and exponential fit. The exponential law fit (Fig. 9B, short dashed line)




includes all data points (including extrapolated data within 100 m of source rock) and
highlights the importance of slope and initial impact velocity at the cliff base, which causes
more boulders to be deposited at greater distances and creates a deviation from the power law
fit (Fig. 9B,  solid line). The exponential fit for CES rockfall boulders is poor and indicates
there is no discernable correlation between CES boulder frequency and runout distance (Fig.
9B, long dashed line). Despite the low number of CES boulders (n=48), it is interesting that
the CES runout distribution shows such a noticeable deviation from the pre-CES data set and
could reflect the influence of deforestation on runout distance. This would imply that the
incremental input of CES and future rockfalls at Rapaki (emplaced during bare-earth
conditions) will modify the overall trend for the cumulative rockfall data set.

At Rapaki, the shadow-angle $Ht/Lt$ relationship is fit best using a polynomial regression (Fig.
9C). The trend indicates a positive correlation between talus slope height (Ht) and travel
distance on the talus slope (Lt), with a reduction in the rate of increase as rockfall runout (Lt)
increases. At Purau, CES and pre-CES rockfalls (within individual geomorphic domains) are
modeled using a variety of data functions (e.g. linear, log, polynomial), suggesting intra-site
geomorphic and geologic factors affecting rockfall hazard are spatially variable (Fig. 9D). We
note that Copons (2009) reports linear regression lines for historical rockfalls in the Central
Pyrenees using the shadow-angle method, and locally, Massey et al. (2014) also show linear
regression fits using the shadow-angle method for CES rockfalls in the Port Hills of southern
Christchurch. Our data indicates that non-linear regression functions (for the shadow-angle
method) are more successful in capturing the Ht/Lt relationship as distance from the source
rock increases.

No clear relationship is obtained between boulder volume and runout distance at Rapaki (Fig.
9E) and Purau (Fig. 9F). At both sites, a wide range of boulder sizes exist for the full spectrum
of pre-CES and CES $Ht/Lt$ ratios, suggesting that boulder size  is not a primary driver for runout
distance at the study sites; although it is possible that smaller boulders (e.g. ~1-2 m$^3$) exhibiting
long runout distances (i.e. low $Ht/Lt$ ratios) may represent smaller rock fragments detached
from larger boulders during transport and eventual emplacement on the hillslopes and within
valleys.

**5.6  RAMMS rockfall modelling**



### 5.6.1 RAMMS_1

A primary challenge in replicating the distribution of CES rockfalls was determining an appropriate set of terrain parameters for the drainage valleys (see Appendix 1, Table A1). To match the RAMMS boulders with the field-mapped CES rockfalls (Fig. 10A) it was necessary to create separate valley terrain polygons and modify the terrain parameters to reflect the high degree of impedance and/or dampening (Vick et al., 2019) in the drainage gullies (see Appendix 2, Table A1). Our field observations confirm the presence of abundant pre-existing boulders within drainage valleys (Fig. 12A-F) and many instances where CES boulders were stopped by pre-CES rockfalls (see Fig. 12A-C). The effect of pre-CES rockfall debris on boulder transport and final resting location needs to be further investigated in order to effectively model impediments within drainage valleys. Further, a more refined understanding for the influence that substrate soil moisture content has on rockfall runout is required (Vick et al., 2019). We note that the DEM used for our study has a resolution of 4 m and may not adequately simulate the smaller scale surface roughness (e.g. clustering of boulders below this size threshold) observed during our field studies (Fig. 12A-G).

### 5.6.2 RAMMS_2

The RAMMS_2 model scenario (prehistoric/forested hillslope) is moderately successful (slight overprediction) in replicating the overall spatial distribution (including maximum runout distances) for the majority of mapped pre-CES rockfalls at Purau (Figs. 10B). The best fit occurs when the forest density is increased to 10000 kg/s (dense vegetation) and applied uniformly across the Purau hillslopes (see Figure 10B, white dashed line). This represents an increase compared with the forest density used at Rapaki (i.e. 3000 kg/s for moderate vegetation [interfluves], 6000 kg/s for dense vegetation [valleys] (see Borella et al., 2016a) and implies that vegetation may have been denser on the northwest-facing Purau hillslopes compared with the south/southeast facing Rapaki hillslope.

We note the difference between maximum runout distance for RAMMS and empirical pre-CES boulders at the CES-7 site (Fig. 10B). RAMMS predicts that pre-CES boulders should be deposited further from the source rock (maximum runout distance=~325 m) than is observed (maximum runout distance=~105 m) in the field. Several possible explanations exist including: (1) pre-CES boulders were in fact deposited further from the source rock and were


subsequently buried by loess and hillslope colluvium; (2) RAMMS underestimates the effect
of hillslope vegetation during prehistoric times; (3) during pre-CES times less of the source
rock was exposed (due to burial) and therefore the volcanic rock was less susceptible to
detachment during shaking; and/or (4) during pre-CES shaking events the direction of strong
ground shaking was not favourable to rockfall detachment. (5) Scenario 1 is possible but would
need to be confirmed through subsurface trenching or ground penetrating radar (GPR)
methods. Tunnel gulley erosion has exposed sections of the subsurface on the CES-7 hillslope
and no buried boulders are evident. Scenario 2 is probable based on our observations of forested
hillslopes elsewhere in the Port Hills and greater Banks Peninsula area. It is common for dense
native vegetation to grow up to, and in some cases, onto portions of the volcanic source rock.
In these cases, a high volume of detached rockfalls are stopped adjacent to the source rock and
never generate the required momentum to runout an appreciable distance. Scenario 3 is also a
possibility and requires that the CES-7 source rock was partially buried during emplacement
of the pre-CES rockfalls. The last phase of hillslope aggradation would have occurred during
the last glacial maximum (~18-24 ka) and possibly up to ~12-13 ka (see Borella et al., 2016b).
We assume the Purau hillslopes have been net erosional (i.e. downwasting) since the early
Holocene; a condition that would have been significantly accelerated after deforestation in the
Purau area. Option 4 is a final possibility but would require that the ~north facing PD1 source
rock is oriented in such a way that strong ground motions from multiple prehistoric shaking
events were unable to create rockfall detachment to the degree evident in the CES (see section
5.7 for more discussion on strong ground motions).

RAMMS 2 model scenarios effectively identify pre-CES rockfall boulders that have likely
experience post-emplacement mobility (Fig. 10B). This is shown by the collection of pre-CES
boulders within the central drainage canyon that exceed the limit of simulated RAMMS
boulders (Fig. 10B), indicating a transport mechanism other than rockfall. Field observations
confirm that the depositional patterns of boulders located beyond the limits of what RAMMS
predicts are consistent with debris flow and other transport/deposition processes. This is further
highlighted by the numerous and large pre-CES rafted boulders (maximum volume=20 m$^3$)
identified near the Purau coastline (see Fig. 3).

Finally, we observe no mapped pre-CES boulders outside of the valleys that exceed the
RAMMS_2 maximum runout distance (Fig. 10B), implying that the mapped pre-existing
boulders (yellow circles) were deposited prior to deforestation of the Purau hillslopes and are


prehistoric (i.e. deposited prior to European arrival) in age. This result is consistent with
prehistoric boulder ages determined at the Rapaki study site where the youngest emplacement
ages for pre-CES boulders are ~2-6 ka (Mackey and Quigley, 2014; Borella et al., 2016b).

**5.6.3 RAMMS_3**

With the exception of area CES-7, RAMMS_3 maximum runout distances are well in
exceedance of the mapped locations for the CES rockfall boulders (Fig. 10C), and highlights
the potential increased rockfall hazard resulting from input from additional detachment sites,
particularly those overlying hillslopes where boulder trajectories are not as strongly influenced
(i.e. captured) by nearby valleys. The results indicate that development at S1 and S2 sites could
be adversely impacted by future rockfall events (Fig. 10C). Assuming terrain characteristics
remain similar, Sites 3, 4, and 5 are unlikely to be impacted by rockfall boulders in the future,
although additional mapping and related structural studies of the volcanic source rock is
required to determine the most vulnerable rockfall source areas.

**5.7  Interpretations of strong ground motion data**

Preceding studies provide some insight into possible strong ground motion characteristics at
Rapaki and Purau during the Mw 6.0 and 6.2 earthquakes. Kaiser et al.'s (2014) seismic array
analysis of weak ground motion provides information regarding frequency-dependent
amplification at Kinsey Terrace, Redcliffs, and Mt. Pleasant (henceforth Ksites), all of which
are north-facing slopes in the Port Hills. They found that both morphological features as well
as properties of the wave propagation media control frequency-dependent amplification. In
particular, significant ground motion amplification was observed at $1 - 3$ Hz frequency range
on top of narrow, steep-sided ridges. At these low frequencies (f), seismic wavelengths (λ) are
comparable to ridge width of Ksites. Therefore, seismic waves in the $1 - 3$ Hz frequency band
appear to excite natural resonance (or natural frequency; $f_n$), optimizing ground motion.

It is interesting to evaluate the implications of Kaiser et al.'s (2014) low frequency observations
to Rapaki and Purau rockfall sites. Both these sites are located at higher elevations than Ksites.
Thus, their ridge width (~400 – 500 m) is somewhat less than that at Ksites (~ 600 – 1000 m).
Using this information, we estimate $f_n$ to be < 5 Hz (see supp infor).





Whether ground motion with $f_n$ was excited at these sites depends on the amount of energy
carried by seismic waves in that frequency band. This information is contained in the spectra
of velocity seismograms – a proxy for kinetic energy distribution over frequency. We selected
D13C station for this preliminary analysis because the distance between this station and the
Rapaki site is only about 2 km. They are also at similar elevations with ridge morphologies
resembling each other. Rapid variations in geological conditions are unlikely over such short
length-scales, which allows us to extrapolate both high and low frequency wave characteristics
observed at D13C station to Rapaki with less uncertainty than the other stations. The nearest
station to Purau is LPCC (~ 5 km). The two sites are vastly different as LPCC is located at the
toe of a steep cliff in the Lyttelton Port, whereas Purau sites are high elevation ridges. Thus,
ground motion recorded at LPCC is not a reliable proxy for ground motion characteristics at
Purau. The next nearest station D15C is ~ 7 km from Purau and it suffers from morphological
dissimilarities (variations in ridgeline orientation and morphology) that make extrapolating
ground motion between the sites highly unreliable. Despite the fact that D13C station is located
~10 km from Purau, similarity of morphological features including elevation makes D13C a
desirable station to understand ground motion at Purau.

We computed velocity spectra of east and north components of the station D13C (Fig. 13) to
qualitatively assess seismic energy transmission through our rockfall sites. We find that the
transition from the flat spectrum to a rapid fall off occurs at ~3 – 4 Hz. This means that the
2011-06-13 Mw 6 earthquake carried most of its energy at frequencies less than ~3 – 4 Hz.
Together with our estimates of $f_n$ (< 5 Hz), we can thus infer that the passage of seismic waves
excited natural resonance at Rapaki and Purau sites. The combined effects of natural resonance
and wave focusing towards the ridge crest (Hartzell et al., 1994; Bouchon & Barker, 1996) in
these hard rock sites have the potential to optimize shaking, promoting rockfalls.

It is interesting to note, however, that D13C recorded the lowest peak velocities (223 mm/s and
178 mm/s) and displacements (38 mm and 74 mm) of the four stations considered here (Fig.
7C). Out of these stations, it is also the only station that recorded no acceleration above 0.3g
on any component. These features of the wavefield are not surprising because distance from
D13 C to epicentre of the Mw 6 earthquake is twice (~9 km) as large as that from the other
stations (~4.5 km). For this reason, it is likely that other possible effects (e.g., rockmass
weakening by prior CES earthquakes), in addition to strong ground motions from the Mw 6



earthquake, were responsible for triggering major rockfalls at the study sites. Unfortunately,
D13C was not in operation at the time of these previous larger earthquakes to assess severity
of ground motion. Nonetheless, records from stations closest to D13C indicate that those sites
have exceeded the 0.3g peak ground acceleration (PGA) threshold important for engineering
considerations. For instance, LPCC station located ~6 km from D13C recorded 0.3g and 0.9g
PGA following the Mw 7.1 and Mw 6.2 events respectively (Bradley & Cubrinovski, 2011).
Moreover, extrapolation of PGA contours of Bradley (2012) suggests that D13C and Rapaki
sites experienced  PGA exceeding 0.25g and 0.45g during Mw 7.1 and Mw 6.2 earthquakes
respectively. Some of the rockfall sites investigated herein might have had reached a critical
failure threshold prior to being triggered by the 2011-06-13 Mw 6 earthquake.

The particle velocity and motion hodograms (Fig. 7A-E) we computed also carry directional
information of particle behaviour in addition to intensity that we discussed earlier. Past studies
show that seismic wave polarizations are amplified in directions perpendicular to fracture
surfaces, weakening the coherence between outer blocks of rock with bedrock during the
passage of a seismic wave (Kleinbrod et al., 2017; Burjánek et al., 2018). If blocks of rock are
primed for failure by previous events, this effect can produce rockfalls at a local magnitude as
small as ~4 (Keefer, 1984). The velocity hodogram of D13C exhibits a strong ENE-WSW
component. Note that this direction makes roughly ~30º to ~60º angle with rock faces at PD2,
PD3, PD4, and RAP sites (Fig. 7C). Thus, it is reasonable to assume that particle velocities in
this dominant direction are favourable for triggering rockfalls particularly if the rock faces were
primed for failure. The angle between this dominant velocity component and the rock face at
PD1 site, however, appears to be less than ~20º and possibly is not as favourable for triggering
rockfalls as for other sites. On the other hand, the particle motion hodogram has two dominant
directions; WNW and WSW. Depending on the strike of the rock face, either one of these
directions can orient particle motion favourably for rockfalls. For instance, site RAP has a rock
face strike of 25º, which is sub-parallel to the WSW particle motion direction. However, the
WNW particle motion direction makes a steep angle with the rock face and thus can promote
rockfalls. Combining information from particle velocity and motion hodograms, we
hypothesize that directional aspects were favourable to rockfall triggering at the Rapaki and
Purau sites.
**5.8  Pre-existing rockfalls as predictive database**


Our study indicates that pre-CES rockfalls provide an accurate range of expected boulder
volumes, shapes, and % lithologic variance (i.e. VB vs CL), but underestimates expected
average and maximum runout distances (on interfluves) because pre-CES rockfalls were
probably emplaced on a forested hillslope. Conversely, the final resting locations for pre-CES
boulders in well-established drainage valleys/channels may overestimate the expected runout
for future rockfalls because the rockfalls have been remobilized after their initial emplacement.

Prior to the CES, rockfall hazard was not considered a high threat in Banks Peninsula and
surrounding areas (Townsend and Rosser, 2012), including the Port Hills of southern
Christchurch, where damage was most critical and 5 fatalities occurred (Massey et al., 2014).
To date, we are aware of only four studies that have dated pre-CES rockfalls in Banks Peninsula
(Mackey and Quigley, 2014; Borella et al., 2016b, Sohbati et al., 2016; Litchfield et al., 2016),
and all of these investigations occurred after the CES. We assume this was primarily because
there were few records of historical rockfall occurrence, and of those described (Lundy, 1995),
none hinted at the potential for future widespread cliff collapse and rockfall in the region.
However, the geologic record (i.e. prehistoric rockfalls) provides evidence that rockfall events
of similar magnitude (or greater) have occurred in the past. In regions devoid of historical or
contemporary rockfalls, pre-existing rockfalls represent the only empirical proxy for evaluating
local rockfall behavior and provide valuable input for rockfall modeling and risk assessment
studies. Existing rockfalls provide valuable data for predicting rockfall volumetric, lithologic,
and morphologic (i.e. boulder shape) characteristics, but a thorough consideration of landscape
evolutionary chronologies (including deforestation) and post-emplacement mobility scenarios
is required before pre-existing rockfalls can be confidently used as future spatial indicators.

**6 Conclusions**

The spatial distributions and physical-geological properties of individual (n=1093) rockfall
boulders deposited at two sites in Banks Peninsula prior to the 2010-2011 Canterbury
earthquake sequence (CES) are compared to boulders (n=185) deposited during the CES. Pre-
CES to CES boulder ratios range between 5:1 and 8.5:1 respectively, suggesting preservation
of multiple pre-CES rockfall events with a flux analogous to or smaller than CES events, and
/ or pre-CES event(s) of larger flux. Pre-CES and CES boulders at one site (Purau site) have
statistically-consistent power-law frequency-volume distributions between 1.0 to >100.0 $m^3$.
At the Rapaki site, CES boulders have smaller and more clustered volumetric distributions that




are less well fit by power-laws compared with the pre-CES data, interpreted to reflect variations
in rockfall source characteristics through time. Boulders of volcanic breccia (VB) have a larger
binned-percentage of large volume boulders and more equant boulder aspects relative to
coherent lava (CL) boulder lithologies at both sites, revealing lithologic controls on rockfall
physical properties. The maximum runout distances for Rapaki CES VB and CL boulders are
greater than that of pre-CES boulders of equivalent lithologies, volumes and morphologies.
This is interpreted as an effect of anthropogenic deforestation on the hosting hillslope, which
enabled CES boulders to travel further than their pre-CES counterparts due to reduced
resistance from vegetation. At Purau, isolated geomorphic domains exhibit this same effect,
however in other intra-site locations, pre-CES boulder locations exceed runout distances of
CES boulders. This is interpreted to reflect post-depositional mobility of prehistoric boulders
via debris flows and other surface processes, reduction of CES boulder runouts in channels due
to collisional impedance from pre-CES boulders, and heterogeneity in the CES boulder
distributions, which reduced the likelihood of large runout boulders occurring due to smaller
volumetric fluxes. The shadow angle method is a reliable predictor for pre-CES and CES
rockfall runout at both sites. At Rapaki, the pre-CES and CES rockfall data is best fit using a
$2^{nd}$ order polynomial regression, while at Purau rockfalls require a variety of data fits (e.g.
linear, log, polynomial), suggesting intra-site geomorphic and geologic factors affecting
rockfall hazard are spatially variable. Bare-earth and forested numerical modeling suggest that
the majority of pre-CES rockfalls were emplaced before deforestation of the Purau hillslopes
and enables identification of boulder sub-populations that have likely experienced post-
emplacement mobility. Our study highlights the challenges of using rockfall distributions to
characterize future rockfall hazards in the context of geologic and geomorphic variations,
including natural and anthropogenically-influenced landscape changes.

*Acknowledgements.*   Financial support for the project came from the EQC (Earthquake
Commission) capability fund for South Island geohazards research. J.B. thanks Sarah Trutner,
Peter Borella, Maxwell Borella, Peter Almond, Simon Brocklehurst, David Bell, and Jarg
Pettinga. Special thanks to Pip and David Barker for allowing us land access in Purau and
review of the Camp Bay geotechnical property report. The authors declare that they have no
competing interests.





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





Natural Hazards and Earth System Sciences Discussions — Open Access — EGU

| | Rapaki Pre-CES (n=409) (m³) | Rapaki CES (n=48) (m³) | Difference (m³) | Difference (%) | Purau Pre-CES (n=684) (m³) | Purau CES (n=136) (m³) | Difference (m³) | Difference (%) |
|---|---|---|---|---|---|---|---|---|
| 25th (Q1) | 1.60 | 1.36 | 0.24 | 17.65 | 1.42 | 1.34 | 0.08 | 5.97 |
| Median | 2.94 | 2.21 | 0.73 | 33.03 | 2.20 | 2.01 | 0.19 | 9.45 |
| 75th (Q3) | 6.59 | 4.83 | 1.76 | 36.44 | 5.08 | 4.46 | 0.62 | 13.90 |
| 95th | 20.54 | 19.76 | 0.78 | 3.95 | 27.06 | 17.66 | 9.4 | 53.23 |
| Maximum | 200.56 | 28.35 | 172.21 | 607.44 | 616.00 | 79.97 | 536.03 | 670.29 |
| Mean | 6.81 | 4.84 | 1.97 | 40.70 | 8.10 | 5.32 | 2.78 | 52.26 |

**Table 1.** Volumetric comparison of pre-CES and CES rockfall boulders (for volume ≥1.0 m³) at Rapaki and Purau study sites.

| | *Rapaki* | | | | *Purau* | | | |
|---|---|---|---|---|---|---|---|---|
| | Pre-CES | CES | Pre-CES | CES | Pre-CES | CES | Pre-CES | CES |
| | VB (n=391) (m³) | VB (n=45) (m³) | CL (n=18) (m³) | CL (n=3) (m³) | VB (n=436) (m³) | VB (n=127) (m³) | CL (n=248) (m³) | CL (n=9) (m³) |
| 25th (Q1) | 1.68 | 1.39 | 1.22 | 1.03 | 1.70 | 1.36 | 1.20 | 1.13 |
| Median | 3.1 | 2.21 | 1.38 | 1.06 | 3.21 | 2.04 | 1.56 | 1.68 |
| 75th (Q3) | 6.78 | 5.7 | 1.54 | 1.67 | 7.65 | 4.87 | 2.30 | 2.14 |
| 95th | 21.28 | 20.576 | 3.92 | 2.16 | 40.91 | 17.78 | 5.26 | 2.48 |
| Maximum | 200.56 | 28.35 | 9.99 | 2.28 | 616.00 | 79.97 | 26.21 | 2.64 |
| Mean | 7.03 | 5.06 | 1.96 | 1.45 | 11.43 | 5.58 | 2.24 | 1.67 |
| Total volume | 2749.07 | 227.80 | 35.29 | 4.34 | 4938.76 | 708.34 | 555.63 | 15.00 |
| % of total volume | 99 | 98 | 1 | 2 | 89 | 98 | 11 | 2 |
| % of mapped boulders | 96 | 94 | 4 | 6 | 64 | 93 | 36 | 7 |

**Table 2.** Comparison of boulder size statistics for Rapaki and Purau VB and CL pre-CES and CES rockfall boulders (volume ≥1.0 m³).




| | # of pre-CES rockfalls : # of CES rockfalls (n) | pre-CES : CES ratio | pre-CES : CES % : % | volume of pre-CES rockfalls : volume of CES rockfalls (m³) | pre-CES : CES ratio | pre-CES : CES % : % |
|---|---|---|---|---|---|---|
| Total (Rapaki + Purau) | 1093 : 184 | 5.94 | 86 : 14 | 8323.76 : 955.48 | 8.71 | 90 : 10 |
| Rapaki Total | 409 : 48 | 8.52 | 89 : 11 | 2784.37 : 232.14 | 11.99 | 92 : 8 |
| Rapaki VB | 391 : 45 | 8.69 | 90 : 10 | 2749.07 : 227.80 | 12.07 | 92 : 8 |
| Rapaki CL | 18 : 3 | 6.00 | 86 : 14 | 35.29 : 4.34 | 8.14 | 89 : 11 |
| Purau Total | 684 : 136 | 5.03 | 83 : 17 | 5539.39 : 723.35 | 7.66 | 88 : 12 |
| Purau VB | 436 : 127 | 3.43 | 77 : 23 | 4983.76 : 708.34 | 7.04 | 88 : 12 |
| Purau CL | 248 : 9 | 27.56 | 96 : 4 | 555.63 : 15.00 | 37.04 | 97 : 3 |

**Table 3A.** Comparison of frequency (n) and volume (m³) ratios for pre-CES and CES rockfall boulders at the Rapaki and Purau study sites.

| | # of VB boulders : # of CL boulders (n : n) | VB : CL ratio | VB : CL % : % | Volume of VB boulders : volume of CL boulders (m³ : m³) | VB:CL ratio | VB:CL % : % |
|---|---|---|---|---|---|---|
| Total (Rap + Purau) | 999 : 278 | 3.59 | 78 : 22 | 8668.97 : 610.26 | 14.21 | 93 : 7 |
| Rapaki Total (pre-CES + CES) | 436 : 21 | 20.76 | 95 : 5 | 2976.87 : 39.63 | 75.11 | 99 : 1 |
| Rapaki pre-CES | 391 : 18 | 21.72 | 96 : 4 | 2749.07 : 35.29 | 77.9 | 99 : 1 |
| Rapaki CES | 45 : 3 | 15 | 94 : 6 | 227.80 : 4.34 | 52.49 | 98 : 2 |
| Purau Total (pre-CES + CES) | 563 : 257 | 2.19 | 69 : 31 | 5692.1 : 570.63 | 9.98 | 91 : 9 |
| Purau pre-CES | 436 : 248 | 1.76 | 64 : 36 | 4983.76 : 555.63 | 8.97 | 90 : 10 |
| Purau CES | 127 : 9 | 14 | 93 : 7 | 708.34 : 15.00 | 47.22 | 98 : 2 |
| Purau D1 pre-CES | 17 : 0 | N/A | 100 : 0 | 137.27 : 0 | N/A | 100 : 0 |
| Purau D1 CES | 30 : 0 | N/A | 100 : 0 | 125.86 : 0 | N/A | 100 : 0 |
| Purau D2 pre-CES | 36 : 3 | 12 | 92 : 8 | 230.8 : 3.9 | 59.18 | 98 : 2 |
| Purau D2 CES | 1 : 1 | 1 | 50 : 50 | 14.78 : 1.08 | 13.69 | 93 : 7 |
| Purau D3 pre-CES | 54 : 43 | 1.26 | 56 : 44 | 203.79 : 142.62 | 1.43 | 59 : 41 |
| Purau D3 CES | 38 : 3 | 12.67 | 93 : 7 | 242.63 : 5.91 | 41.05 | 98 : 2 |
| Purau D4 pre-CES | 8 : 1 | 8 | 89 : 11 | 188.42 : 1.24 | 151.95 | 99 : 1 |
| Purau D4 CES | 36 : 0 | N/A | 100 : 0 | 267.76 : 0 | N/A | 100 : 0 |

**Table 3B** Comparison of VB/CL frequency (n) and volume (m³) ratios for pre-CES and CES rockfall boulders at the Rapaki and Purau study sites.






| Runout Distance (MLR) | Average (m) | Maximum (m) |
|---|---|---|
| **Rapaki** | | |
| Pre-CES | 184.30 | 567.51 |
| CES | 276.23 | 702.47 |
| *Pre-CES VB* | *184.65* | *567.51* |
| *Pre-CES CL* | *176.57* | *346.73* |
| *CES VB* | *276.91* | *702.47* |
| *CES CL* | *266.13* | *432.14* |
| **Purau** | | |
| PD1 Pre-CES | 29.86 | 96.96 |
| PD1 CES | 119.63 | 348.4 |
| PD2 Pre-CES | 84.01 | 279.75 |
| PD2 CES | 14.11 | 15.91 |
| PD3 Pre-CES | 239.62 | 462.8 |
| PD3 CES | 237.24 | 413.35 |
| PD4 Pre-CES | 109.11 | 208.85 |
| PD4 CES | 181.75 | 304.56 |
| *PD1 Pre-CES VB* | *29.86* | *96.96* |
| *PD1 CES VB* | *119.63* | *348.4* |
| *PD1 Pre-CES CL* | *N/A* | *N/A* |
| *PD1 CES CL* | *N/A* | *N/A* |
| *PD2 Pre-CES VB* | *88.73* | *279.75* |
| *PD2 CES VB* | *12.3* | *12.3* |
| *PD2 Pre-CES CL* | *27.39* | *33.38* |
| *PD2 CES CL* | *15.91* | *15.91* |
| *PD3 Pre-CES VB* | *248.96* | *434.85* |
| *PD3 CES VB* | *243.21* | *413.35* |
| *PD3 Pre-CES CL* | *227.89* | *462.8* |
| *PD3 CES CL* | *161.68* | *178.53* |
| *PD4 Pre-CES VB* | *106.99* | *208.85* |
| *PD4 CES VB* | *181.75* | *304.56* |
| *PD4 Pre-CES CL* | *126.06* | *126.06* |
| *PD4 CES CL* | *N/A* | *N/A* |

MLR = Map Length Runout
PD1 = Purau Domain 1

**Table 4**. Average and maximum runout distances for pre-CES and CES rockfall boulders (for volume ≥1.0 m$^3$) at Rapaki and Purau study sites.




**Figure Captions**

**Fig. 1.** **(A)** Google Earth image showing Rapaki and Purau study sites. CES rockfall locations as mapped by GNS Science and the author (at Rapaki and Purau) are shown (red). Epicenter locations for 22 February, 13 June, and 16 April 2011 events are displayed [Modified from Massey et al. (2014)]. Inset map of South Island (New Zealand) shows Banks Peninsula and approximate location for study site (yellow star). **(B)** Anthropogenic deforestation of Banks Peninsula. Removal of native forest occurred rapidly in Banks Peninsula (BP) with arrival of Polynesians (c. AD 1280) then Europeans (c. AD 1830). Before Polynesian (Maori) arrival, extensive native forest was present throughout BP. Prior to European settlement, minor to moderate removal of indigenous forest by Maori occurred, with burning being the primary tool for clearance (yellow). By 1920 Europeans had removed >98% of BP native forest (red). Minor re-establishment of old-growth native forest has occurred (green) but slopes in the Port Hills and greater BP (including Rapaki and Purau) remain largely unvegetated.

**Fig. 2.** **(A)** Mapped pre-CES volcanic breccia (VB) and coherent lava (CL) boulders at Rapaki. The largest boulders with the furthest runout distances are comprised exclusively of volcanic breccia. Ratio of pre-CES VB to CL boulders is ~22:1. **(B)** Mapped CES VB and CL boulders at Rapaki study site. Note the low number of CL rockfall boulders detached during the CES at Rapaki. Ratio of CES VB to CL boulders is 15:1. [a = volcanic source rock; b = dominated by volcanic boulder colluvium and volcanic loess colluvium; c = loess-colluvium underlain by in-situ loess and volcanic rock; d = alluvial sediments overlying loess and bedrock]

**Fig. 3.** **(A)** Mapped pre-CES and CES rockfalls with volume ≥1.0 m$^3$ at Purau study site. Ratio of pre-CES to CES boulders is ~5:1. A= volcanic source rock; B=dominated by volcanic boulder colluvium and volcanic loess colluvium; C=loess-colluvium underlain by in-situ loess and volcanic rock; D=alluvial sediments overlying loess and bedrock. **(B)** Mapped pre-CES VB and CL boulders at Purau. Ratio of pre-CES VB to CL boulders is ~2:1. **(C)** Mapped CES VB and CL boulders at Purau study site. Note the low number of CL rockfall boulders detached during the CES at Purau. Ratio of CES VB to CL boulders is ~14:1. PD1-PD4 represent Purau rockfall domains.

**Fig. 4.** Pre-CES and CES VB boulders at Rapaki and Purau study sites. **(A)** Pre-CES boulder in footslope position with smaller CES boulder at right bottom. **(B)** Exploratory trenching exposes the colluvial sediment wedge at the boulder backside depicted in Fig. 7B. **(C)** Pre-CES boulder at Purau study site. Erosion of the surrounding hillslope sediments has exposed the boulder base and underlying loessic sediment. **(D)** Advanced surface roughness and abundant lichen growth on pre-CES boulder surface. **(E)** Large CES boulder (~28 m$^3$) detached from Mount Rapaki and emplaced in the Rapaki village during the 22 February 2011 earthquake (photo courtesy of D.J.A. Barrell, GNS Science). **(F)** CES boulder showing 2011 detachment surface [1] and adjacent non-detached surface [2] with higher degree of rough. **(G-K)** Representative CL boulders at Rapaki and Purau sites exhibit typical elongate and flat morphologies.

**Fig. 5.** **(A)** Volcanic source rock at Rapaki study site. Sixty (60) individual detachment zones were created during the CES (yellow) and represent ~9% of the total source rock area. The source rock is comprised of ~86% VB and ~14% CL. ~69% and ~31% of the detachments occurred within the VB and CL lithologies, respectively. **(B)** Photo showing


several irregularly shaped CES detachment zones near the top of Mt. Rapaki. **(C)** Photo showing freshly exposed VB and CL layering within the Rapaki source rock. **(D)** Portion of volcanic source rock at Purau showing VB and CL layering. A single CES detachment site is shown at the top of the source rock. Seven (7) individual CES detachment sites were identified at the Purau study site. **(E)** CL and VB layers at the Purau study site. Note the thickness of the CL layer (~5-7 meters) and lack of any CES detachment sites despite the high degree of fracturing and overhanging condition. **(F)** VB and CL layering in Sumner (Christchurch) cliff exposure adjacent to Main Road. Extensive cliff collapse during the CES has revealed multiple lava flows and the distinctive textural differences between the VB and CL lithologies. Note the high density of vertical to subvertical fractures within the CL layers. **(G)** Exposed lava layers adjacent to Main Road in Redcliffs (Christchurch). Note the single-family living residence at top of photo.

**Fig. 6.** Relative locations of stations LPCC, D13C, D15C, and GODS (yellow squares). Also shown are epicentres of 2011-02-21 Mw 6.2 and 2011-06-13 Mw 6 earthquakes (yellow stars) along with Rapaki and Purau sites.

**Fig. 7.** Each panel shows seismic data from LPCC (A and B), D13C (C), D15C (D), and GODS (E) stations. Panels A and B compare ground motion, respectively, for 2011-02-21 Mw 6.2 and 2011-06-13 Mw 6 earthquakes at LPCC station. The left column shows east and north components of the velocity seismogram (blue line) and their respective envelopes (red dashed-line). The particle velocity hodogram (middle column, green line) was determined for a time window ± 5 s (shaded region in the left column) around the peak (red circle) of the east component envelope. The strike of the rock face (black short line segments) and the direction of the free face (red arrows) for sites PD1, PD2, PD3, PD4, and RAP are also illustrated. The particle motion hodogram (grey line) is presented in the right column, where green, yellow, and red segments represent, respectively, time points at which east component, north component, or both components exceed an acceleration of 0.3g. Note that scale of figure axes varies by station particularly for ground motion.

**Fig. 8.** **(A)** Rockfall size distribution as a proportion of boulders less than a given size plotted in log-space for CES and pre-CES rockfalls at Rapaki. **(B)** Rockfall frequency/size distribution for CES and pre-CES rockfalls at Rapaki. **(C)** Rockfall size distribution as a proportion of boulders less than a given size plotted in log-space for CES and pre-CES rockfalls at Purau. **(D)** Rockfall frequency/size distribution for CES and pre-CES rockfalls at Purau. **(E)** Comparison of boulder size distributions for CES and pre-CES VB and CL rockfalls at Rapaki study site. **(F)** Comparison of boulder size distributions for CES and pre-CES VB and CL rockfalls at Purau.

**Fig. 9.** **(A)** Frequency ratio versus volume ratio for pre-CES and CES rockfall boulders. **(B)** Frequency-runout distributions for Rapaki pre-CES and CES boulders. Both power law (without extrapolated data) and exponential fits (all data) are shown for the prehistoric boulder data set. A poor exponential fit is shown for CES rockfalls. **(C)** Plot of travel distance on talus slope (Lt) versus height on talus slope (Ht) with fitted polynomial regression lines for pre-CES and CES rockfalls at Rapaki. **(D)** Plot of Lt versus Ht with fitted linear, log, and polynomial regression lines for pre-CES and CES rockfalls at Purau. Four (4) separated domains (here D1-D4) are defined at Purau to evaluate the shadow angle method. **(E)** Plot of rockfall size (m$^3$) versus tangent of the shadow angle (Ht/Lt) for Rapaki rockfalls. No tendency of the data is evident. **(F)** Plot of rockfall size (m$^3$) versus tangent of the shadow




angle (Ht/Lt) for Purau rockfalls. The tendency for the domain data sets is poor. Values of correlation coefficients are below 0.3.

**Fig. 10.** **(A)** RAMMS_1 shows deposited rocks for simulated CES boulders. Mapped CES boulders (red circles; n=136) are shown for comparison. Boulder densities of 2500 kg/m$^3$ and 3000 kg/m$^3$ are used for VB and CL boulders, respectively. **(B)** Final resting locations for RAMMS_2 rockfalls. RAMMS-2 assumes prehistoric rockfall conditions (i.e. forested hillslope). Mapped prehistoric rockfalls are depicted (yellow circles) for comparison. An increase in forest density to 10,000 kg/s generates the best fit with maximum runout distance (see white dashed line) for mapped prehistoric boulders. **(C)** Final resting locations for RAMMS_3 boulders. RAMMS_3 assumes modern hillslope conditions (i.e. deforested hillslope) and simulates the future potential rockfall hazard at Purau. The modelling indicates that the distribution of future rockfalls could be widespread and more impactful to existing and proposed development than experienced during the CES. Note the increased maximum runout distance for RAMMS_3 boulders compared with RAMMS_2 and the potential future rockfall hazard to development sites S1 and S2.

**Fig. 11.** RAMMS simulated rockfall boulders showing differences in spatial distribution between VB (mostly equant shaped) and CL (predominantly elongate and flat shaped) boulder morphologies at Purau. All simulated boulders assume a volume of 1.0 m$^3$. **(A)** Spatial distribution of simulated VB boulders at Purau CES-7 location. Note the high relative percentage of simulated boulders deposited at the base of the hillslope (~500-600 meters from source rock). **(B)** Spatial distribution of simulated CL boulders at CES-7 location. Note the higher relative percentage of rockfall boulders deposited near the source rock (within ~100 meters from source rock). The simulation highlights the strong influence of boulder shape on runout distance.

**Fig. 12.** CES and pre-CES rockfall boulders within drainage valleys at Rapaki **(A, C)** and Purau **(B, D, E, F)** study locations. Drainage valleys contain a high amount of pre-CES rockfall boulders, which impacts the trajectory/path of CES rockfalls and stops or reduces runout distance.

**Fig. 13.** Velocity spectra for the 2011-06-13 Mw 6 earthquake recorded at station D13C. No path corrections are applied.





**Appendix 1 - Captions**

**Fig. A1.** The total number of boulders with volume $\geq 0.1$ m$^3$ were taken at runout distances of 1-10 m (yellow polygon 1), 30-40 m (yellow polygon 2), 60-70 m (yellow polygon 3), and 100-110 m (yellow polygon 4) from the volcanic source rock to estimate the total number of boulders in areas near the source cliff where conditions were unsafe for continuous mapping. The number of boulders in areas 'b' and 'c' were reduced by factors of 2 and 3, respectively, based upon field observations. The total number of rockfalls boulders for the area (yellow dashed line) was normalized to boulder size of 1.0 m$^3$ using a power law frequency-size distribution (as determined at the Rapaki study location).

**Fig. A2.** Conceptual diagram of hillslope illustrating the source rock cliff and the talus slope. The reach angle (A) and shadow angle (B) are shown. Sketch modified from Hungr (1993), Wieczorek et al. (2008) and Copons et al. (2009).

**Fig. A3.** Final resting locations for RAMMS_2 rockfalls assuming uniform forest density increase of 10,000 kg/s.

**Appendix 2 - Captions**

**Table A1.** Friction parameters chosen for each terrain type in RAMMS.

**Fig. A1.** Polygon shapefiles for runout terrain types.

**Fig. A2.** Polyline shapefiles for RAMMS_1 rockfall source areas.

**Fig. A3.** Polyline shapefiles for RAMMS_2 and RAMMS_3 rockfall source areas.

**Fig. A4** Polygon shapefiles for forest density.



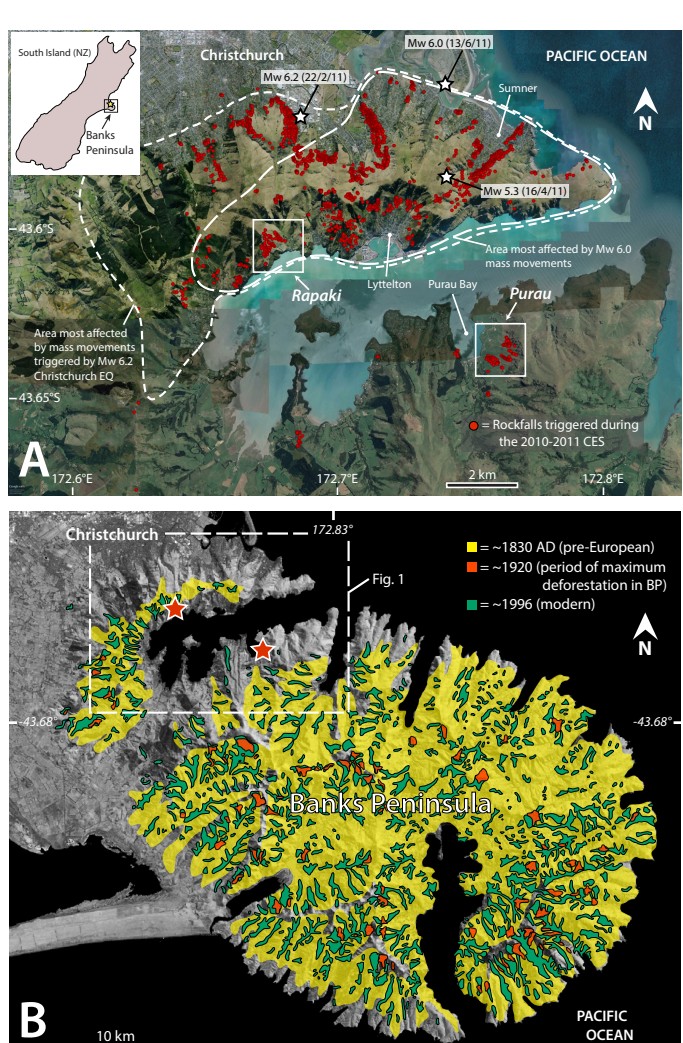

Figure 1


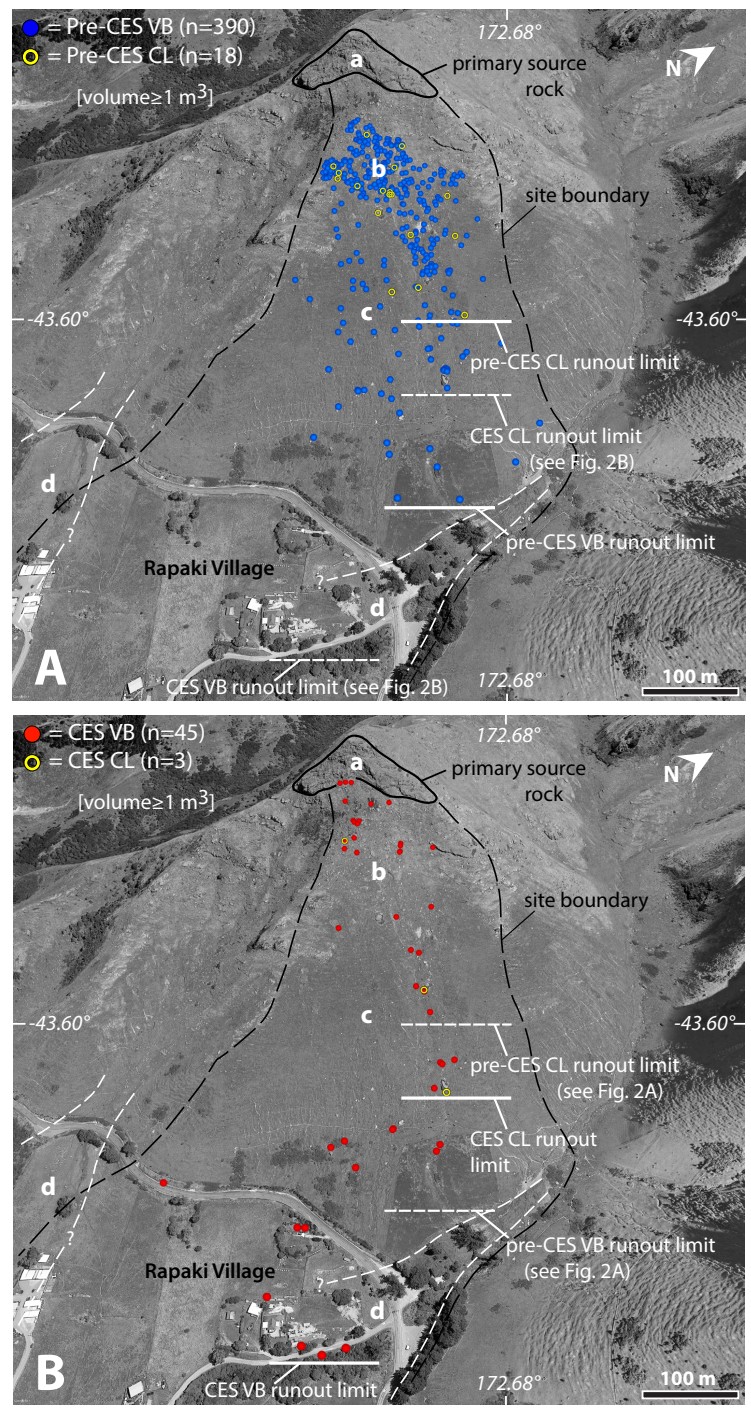

Figure 2


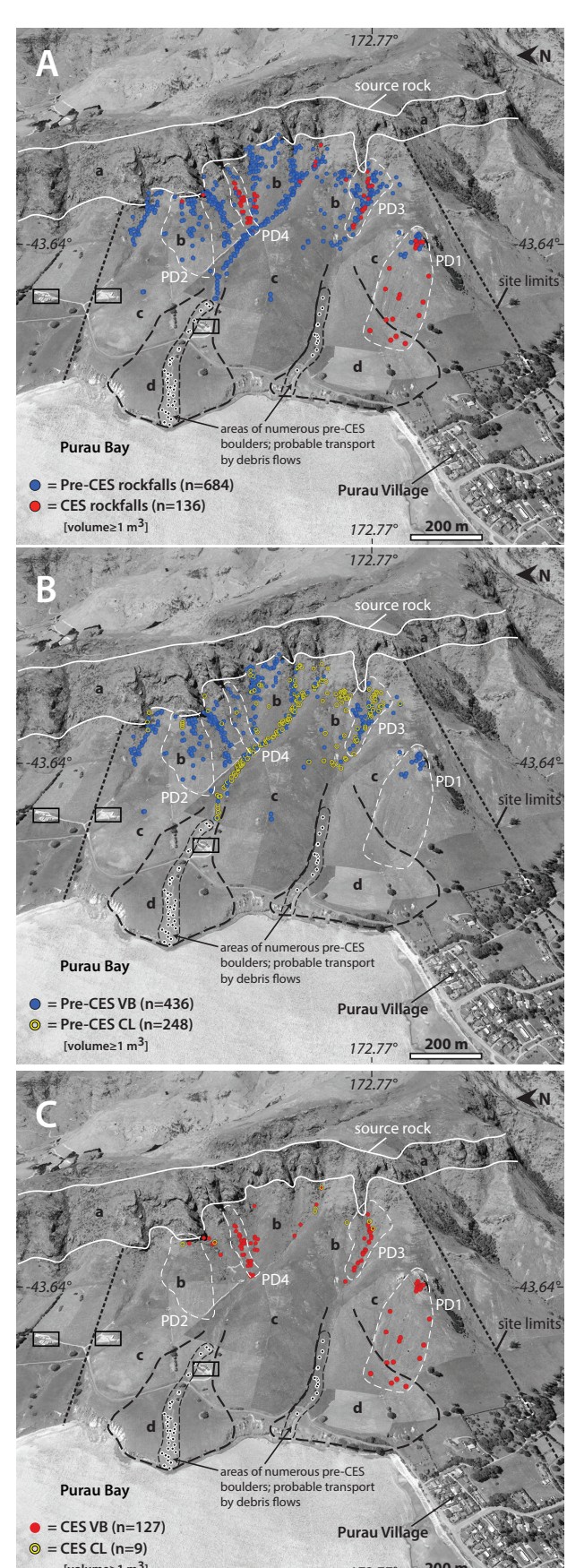

Figure 3




Figure 4




Figure 5





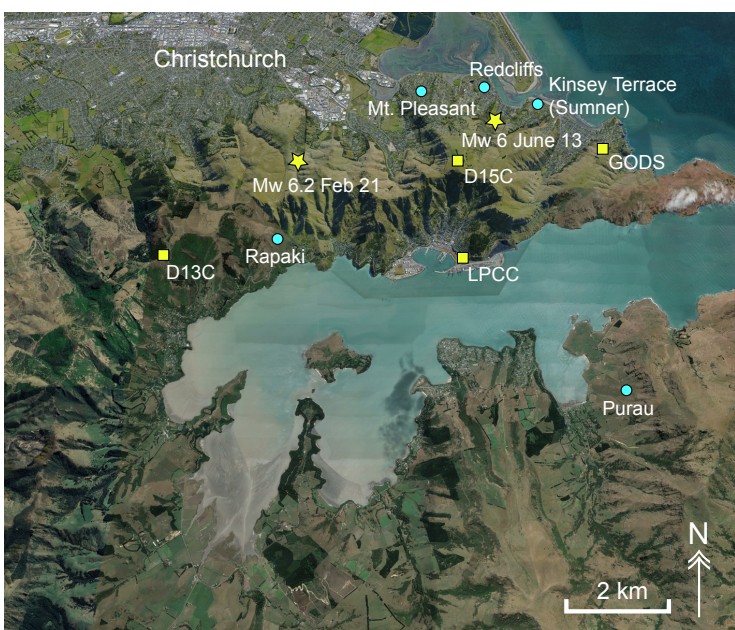

Figure 6


Figure 7





Figure 8




Figure 9



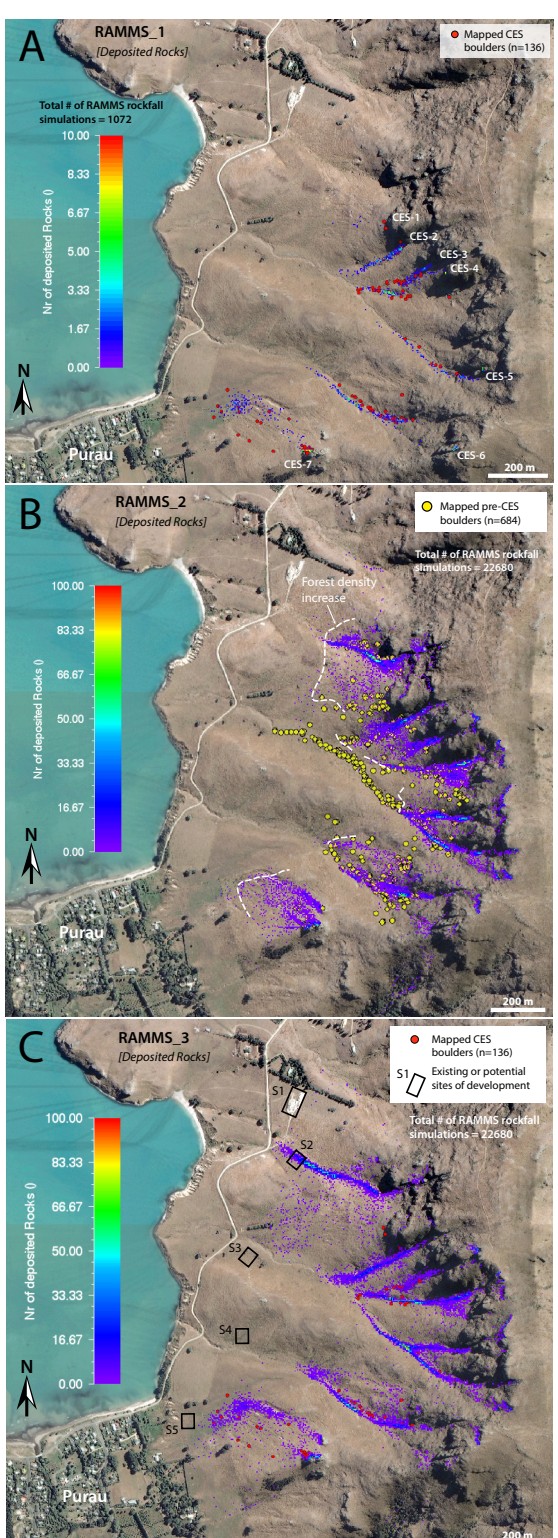

Figure 10




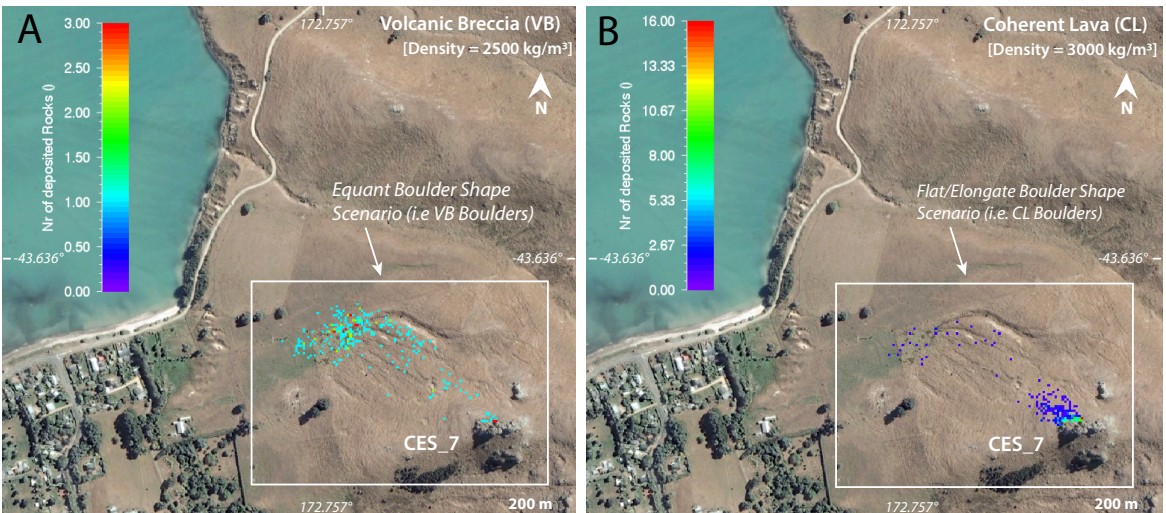

Figure 11





Figure 12



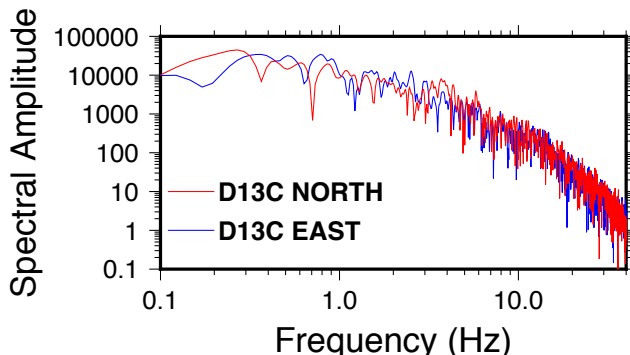

Figure 13





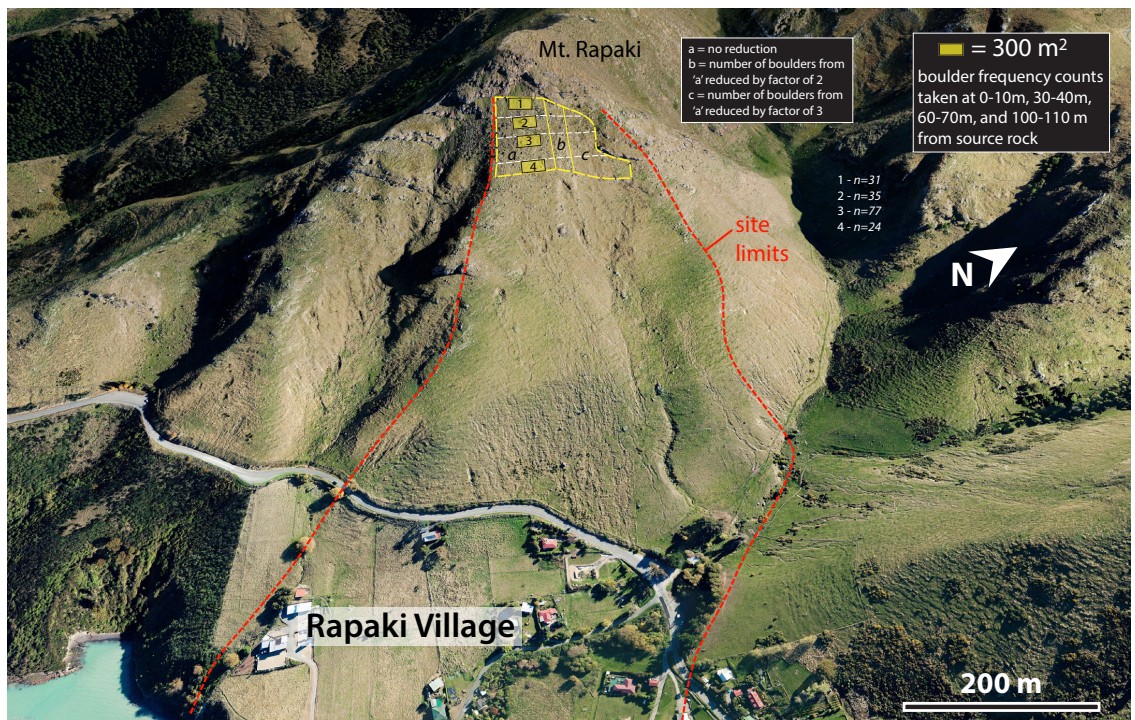

Appendix 1 - Figure A1




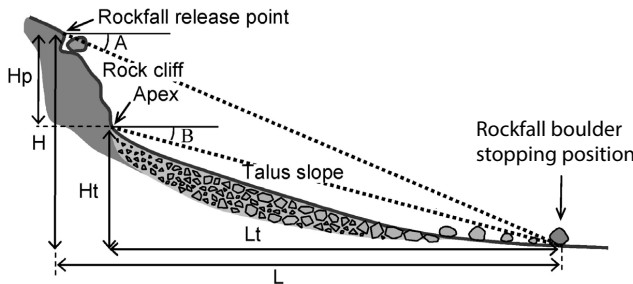

Appendix 1 - Figure A2





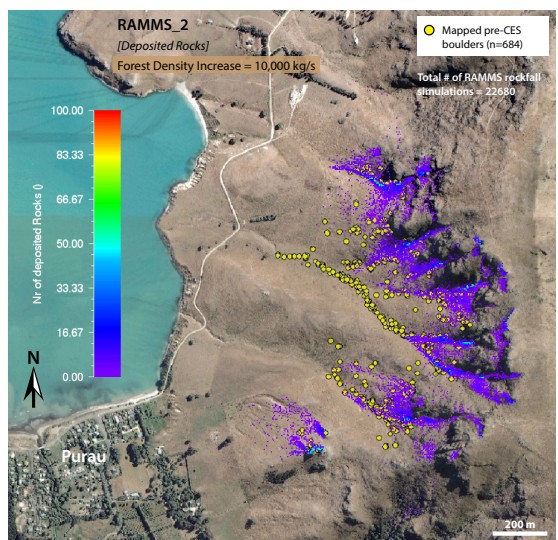

Appendix 1 - Figure A3



| | $\mu_{min}$ | $\mu_{max}$ | $\varepsilon$ | Drag layer coefficient | $\beta$ | $\kappa$ |
|---|---|---|---|---|---|---|
| **Volcanic Rock** | 0.7 | 2.0 | 0 | 0.3 | 50 | 0.5 |
| **Loess and volcanic colluvium** | 0.45 | 2.0 | 0 | 0.5 | 30 | 0.6 |
| **Loess** | 0.3 | 2.0 | 0 | 0.5 | 30 | 0.5 |
| **Valley Terrain** | 0.2 | 2.0 | 0 | 0.9 | 25 | 0.5 |

**Appendix 2_Table A1.** Friction parameters chosen for each terrain type in RAMMS.



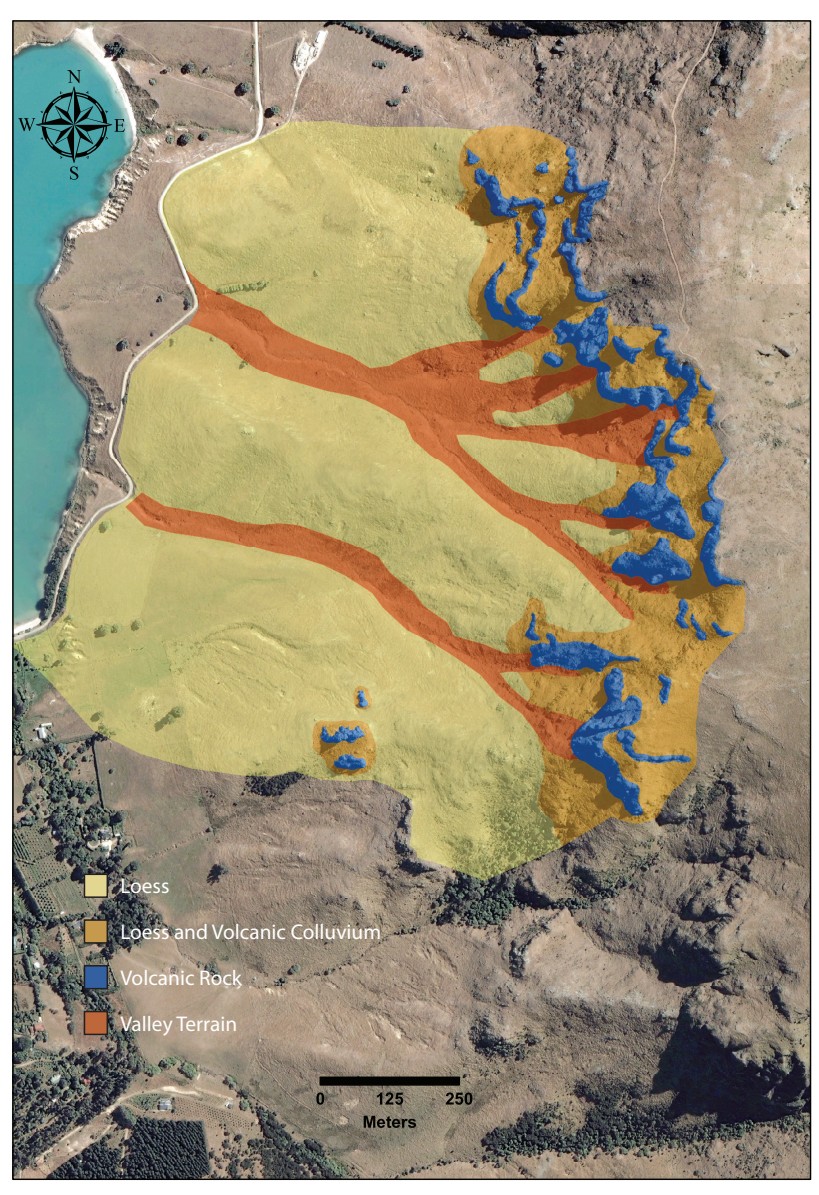


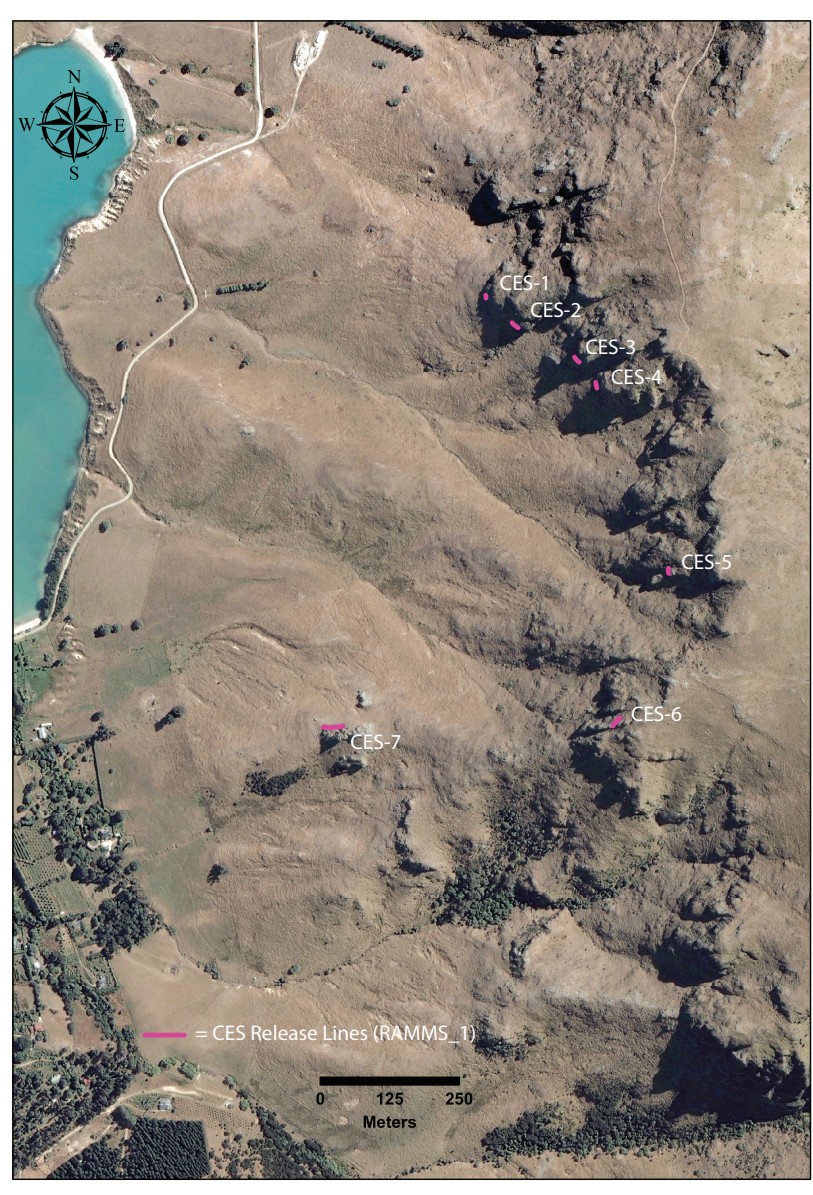


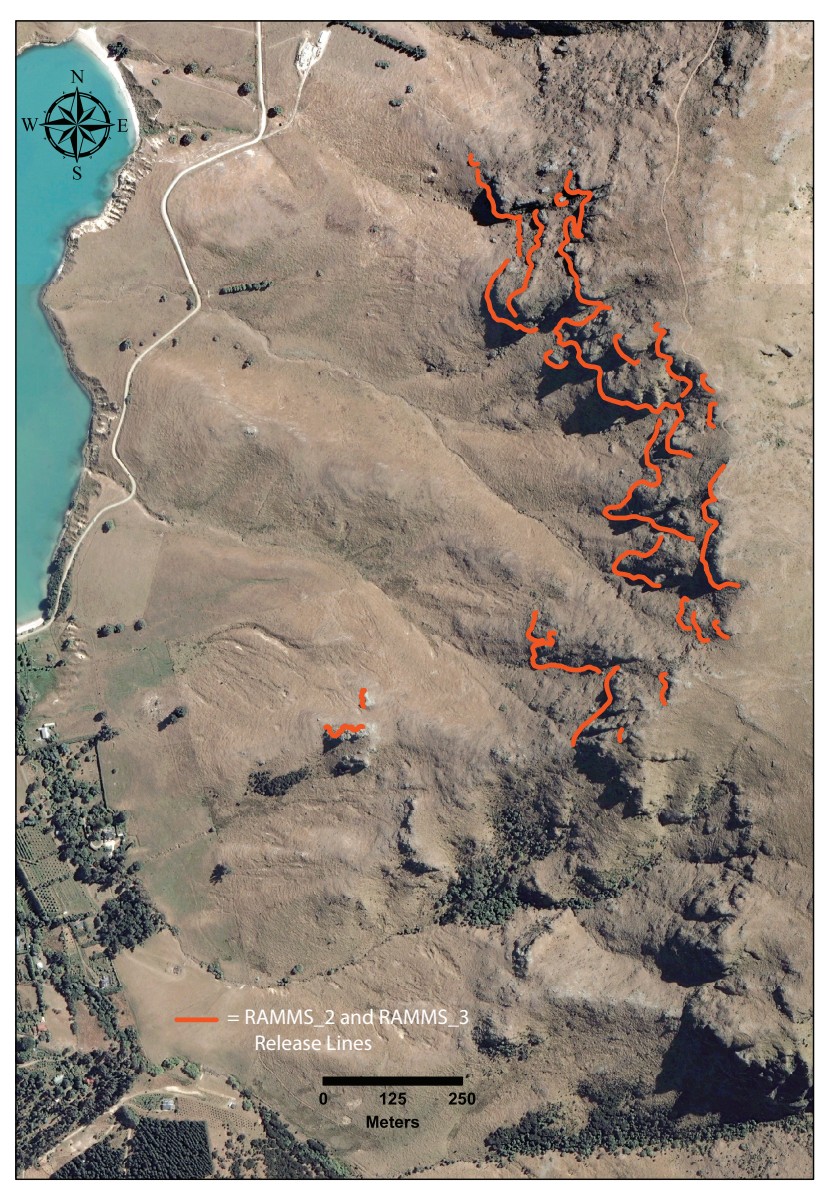



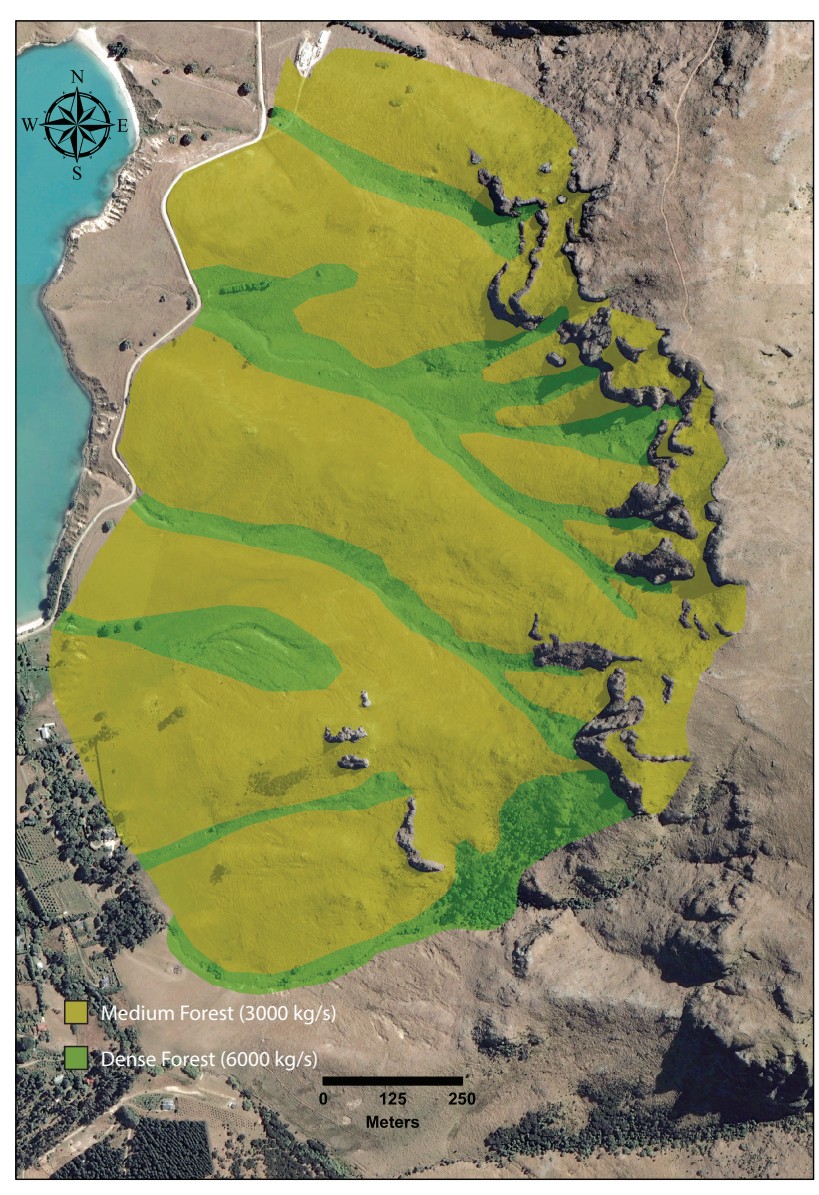