# Peer review of "Geologic and geomorphic controls on rockfall hazard: how well do past rockfalls predict future distributions?"

_Natural Hazards and Earth System Sciences, 2019_

## Referee Comment (RC1) · Martin Mergili (Referee) · 12 Jul 2019

Borella et al. analyze a set of geometric and topographic characteristics of past rockfalls – one set related to the 2010-2011 Christchurch Earthquake Sequence (CES), and one set of older events. Using their measurements and a set of RAMMS simulations, the authors elaborate similarities and differences between the CES and non-CES rock falls, concluding that pre-CES rock falls were generally characterized by lower mobility because slopes were still forested to a higher degree in earlier times than at the time of the CES.

The work addresses an important issue in terms of forecasting of rock fall hazards,

which is certainly within the scope of the journal. The research presented appears sound, and the manuscript is well-written and well-illustrated. Background, methods and results are to a large extent adequately presented and discussed. Before recommending the manuscript for publication, I would like to address some issues and provide some suggestions to the authors to further improve their work. My comments are provided below. In summary, I recommend moderate revisions, but particularly the first issue raised below is critical and important to be addressed adequately.

- An earlier paper led by the same first author (Borella, J. W., Quigley, M., & Vick, L., 2016: Anthropocene rockfalls travel farther than prehistoric predecessors, Science advances, 2(9), e1600969) appears partially similar to the present manuscript in terms of the work described. It should be made clear in this manuscript what are the innovative aspects, compared to the earlier paper.

- Even though I appreciate the very detailed discussion chapter, I have the feeling that there are some redundancies with the results chapter, and some parts of the discussion which might better fit to the results. Consequently, I recommend to revise the results and discussion chapters and to condense the discussion to those issues which are really essential and have not been covered in earlier chapters. This would make it easier for the audience to capture the main points.

- Despite the fact that the manuscript is generally well written, I have found a couple of minor issues of grammar and style – so, please go through the paper carefully again in order to polish the language.

In case the authors would like to discuss the one or the other issue, they should feel free to contact me at martin.mergili@univie.ac.at.

With best regards Martin Mergili

---

## Referee Comment (RC2) · Alexander Preh (Referee) · 5 Aug 2019

This manuscript represents an important contribution to the development of methods for predicting rockfall based on previous (historical) events. The effort involved in the field work and in the preparation of the data is remarkable and without doubt worthy of publication. The fact that there are minor overlaps in content with previous publications has already been noted by the reviewer Mergili.

The conclusions contain clear statements about the possibilities and the quality of the prediction of pre-CES and CES rockfall runout using the shadow angle method (Statement of the authors: The shadow angle method is a reliable predictor). However, there

is no clear statement on how far and in what form the analyses using numerical model RAMMS can be used for predicting of future events. e.g.: Can the Ramms_3 model be used to develop a hazard map? How far is Ramms_3 verified by the models Ramms_2 and Ramms_1? Or is the usefulness of the model calculations limited to the recognition of the effect of deforestation? The authors should supplement the conclusions in this respect, since chapter 5.8 does not contain any specific statements on model calculations either.

In Figures 9c and d, the regression lines are hardly recognizable due to the thick data points. Therefore, it is hardly recognizable to what extent CES and pre-CES differ from each other. This should be corrected.

with kind regrads, Alexander Preh

---

## Author Comment (AC1) · 5 Aug 2019

Dear NHESS,

We appreciate the comments made by referee Dr. Martin Mergili (RC1). Below we respond to each of the interactive comments:

(RC1). An earlier paper led by the same first author (Borella, J. W., Quigley, M., & Vick, L., 2016: Anthropocene rockfalls travel farther than prehistoric predecessors, Science advances, 2(9), e1600969) appears partially similar to the present manuscript in terms of the work described. It should be made clear in this manuscript what are the

[Figure]

innovative aspects, compared to the earlier paper.

JB et al. Response: Our NHESS paper distinguishes itself from Borella et al. (2016) and innovates by investigating a range of influences (i.e. geologic, geomorphic, seismogenic, anthropogenic) on rockfall hazard as they relate to the highly relevant question: How well do past rockfalls predict future distributions? The research is supported by robust prehistoric and contemporary rockfall data sets at two study sites in the Banks Peninsula (NZ), numerical rockfall modelling, and the exceptionally well-recorded seismicity of the 2010-2011 CES. Within our NHESS manuscript we focus on the complexity of interpreting future rockfall hazard based on former boulder distributions due to a variety of natural and anthropogenic factors. [This is different from the motivation behind Borella et al. (2016) which focused primarily on testing the hypothesis that anthropogenic deforestation increases rockfall hazard.] The conditions listed below represent several geological influences comprehensively investigated within our NHESS study that were not within Borella et al. (2016).

- Lithological variability effects on the type of material liberated in successive events and travel path/transport scenario and final resting location.

- Changes in rockfall source (i.e. progressive emergence of bedrock sources from beneath sedimentary cover).

- Remobilization of prior rockfalls by surface processes including debris flows.

- Collisional impedance with pre-existing boulders.

- Variations in location, size, and strong ground motion characteristics of past rockfall-triggering earthquakes and their impact on rockfall flux and boulder mobility.

Our NHESS paper expands upon the Borella et al. (2016) data set by including the Purau study site, which enabled us to evaluate the influences of rockfall hazard over a broader area that included multiple interfluve and drainage canyons.

RAMMS modeling at Purau intentionally used a similar approach/method for evaluating

[Figure]

CES and pre-CES rockfalls. However, there were a few exceptions (see below).

- For Purau we added a RAMMS_3 scenario which models the potential future rockfall hazard at Purau. We assumed bare-earth (deforested) hillslope and dry soil moisture conditions to insure a worst-case (conservative) outcome. Locations for existing and future residential development are shown to highlight the potential impact to dwellings.

- At Purau, separate terrain polygons were defined for drainage valleys. The polygons were assigned a unique set of terrain parameters to account for the influence of collisional impedance with pre-existing boulders and other potential dampening effects within the valleys.

We have made modifications to the NHESS paper to highlight the unique and innovative contributions of JB et al. (2019) NHESS. Please see the attached manuscript (supplemental pdf_JB et al. 2019 Revised Manuscript) below for the applied changes.

(RC1). Even though I appreciate the very detailed discussion chapter, I have the feeling that there are some redundancies with the results chapter, and some parts of the discussion which might better fit to the results. Consequently, I recommend to revise the results and discussion chapters and to condense the discussion to those issues which are really essential and have not been covered in earlier chapters. This would make it easier for the audience to capture the main points.

JB et al. Response: We thank the reviewer for his comments. We've identified several sections within the Discussion that can be included within the Results or removed to avoid redundancy. The changes have helped improve the manuscript. Please see the attached manuscript for the modifications/omissions.

(RC1). Despite the fact that the manuscript is generally well written, I have found a couple of minor issues of grammar and style – so, please go through the paper carefully again in order to polish the language.

JB et al. Response: We have thoroughly reviewed the manuscript and have identified a few locations where grammatical and stylistic errors have occurred. Please see the attached manuscript for the applied changes.

Special note = Modifications and additions to the attached manuscript text are shown in red. Any removed text is shown in red and crossed out.

Please also note the supplement to this comment:
https://www.nat-hazards-earth-syst-sci-discuss.net/nhess-2019-178/nhess-2019-178-AC1-supplement.pdf

**Supplement:**

[revised manuscript text omitted]

---

## Author Comment (AC2) · 15 Aug 2019

Dear NHESS,

We are grateful for the comments made by referee Dr. Alexander Preh (RC2). Below we respond to each of the interactive comments:

(RC2): There is no clear statement on how far and in what form the analyses using numerical model RAMMS can be used for predicting of future events. e.g.: Can the Ramms_3 model be used to develop a hazard map? How far is Ramms_3 verified by the models Ramms_2 and Ramms_1? Or is the usefulness of the model calculations

[Figure]

limited to the recognition of the effect of deforestation? The authors should supplement the conclusions in this respect, since chapter 5.8 does not contain any specific statements on model calculations either.

JB et al. Response: The RAMMS models (in particular, RAMMS_3) have implications for understanding the spatial dimensions of rockfall hazard but are not intended senso stricto to be used as rockfall hazard maps without further site-specific investigations. The primary objective of RAMMS_3 is to show the increased spatial extent (including maximum runout distance) of rockfalls that could result from more widespread source rock detachment (in Purau) under bare-earth (deforested) hillslope conditions. The model does, however, provide a preliminary indicator of low-lying areas (in Purau) that are most susceptible to rockfall hazard and could be used effectively as a means to identify areas that require more in-depth rockfall hazard analyses (which would include an assessment of source rock vulnerability). We recommend that any future rockfall studies using rockfall numerical modeling consider the implementation of boulder morphologies, terrain parameters, and hillslope vegetation attributes developed in this study. We have made additions to the Discussion (Section 5.6.3, lines 817-822) and the Conclusions (Section 6.0, lines 976-981) to address the referee's comments. The additions are presented within the attached revised manuscript and also below:

Discussion (5.6.3) –

'RAMMS_3 highlights the increased spatial extent (including maximum runout distance) of future rockfalls that could result from more widespread detachment within the Purau source rock, particularly for detachment sites overlying hillslopes where boulder trajectories are not as strongly influenced (i.e. captured) by nearby valleys. Although we caution against using RAMMS_3 as a rockfall hazard map, the model results do provide a first-order indicator of low-lying areas that are most susceptible to future rockfall hazard and suggest that development at the S1 and S2 sites could be adversely impacted by future rockfall events (Fig. 10C).'

Conclusions (6.0) –

'The RAMMS_3 model effectively shows the potential spatial extent of rockfalls that could result from more widespread detachment within the Purau volcanic source rock and provides a preliminary indicator of low-lying areas most susceptible to future rockfall hazard. More in-depth rockfall hazard analyses (including numerical rockfall modeling) are required at Purau and should consider the implementation of boulder morphologies, terrain parameters, and hillslope vegetation attributes developed in this study.'

(RC2): In Figures 9c and d, the regression lines are hardly recognizable due to the thick data points. Therefore, it is hardly recognizable to what extent CES and pre-CES differ from each other. This should be corrected.

JB et al. Response: Figures 9C and D have now been modified to ensure that the regression lines are clearly shown and the reader is able to compare/contrast the individual regression lines for the CES and pre-CES data sets. In order to ensure the regression lines are clear to the reader, the CES and pre-CES lines have been colored red and blue, respectively. Further, and for the sake of consistency within Figure 9, we have made the same changes to the regression lines displayed on Figure 9B. The revised Figure 9 is included at the end of the submitted revised manuscript (see supplemental PDF).

Special note: RC2 text changes/additions to the manuscript are colored blue to distinguish from those modifications made to address RC1 comments.

Please also note the supplement to this comment:
https://www.nat-hazards-earth-syst-sci-discuss.net/nhess-2019-178/nhess-2019-178-AC2-supplement.pdf

**Supplement:**

[revised manuscript text omitted]

Figure 9